

# GABLS4 intercomparison of snow models at Dome C in Antarctica.

Patrick Le Moigne[1], Eric Bazile[1], Anning Cheng[3], Emanuel Dutra[4], John M. Edwards[2], William Maurel[1], Irina Sandu[5], Olivier Traullé[8], Etienne Vignon[6], Ayrton Zadra[7], Weizhong Zheng[3]

[1]CNRM, Université de Toulouse, Météo-France, CNRS, Toulouse, France
[2]Met Office, FitzRoy Road, Exeter, UK
[3]IMSG, Inc.@EMC/NCEP/NOAA, College Park, Maryland, United States
[4]Instituto Português do Mar e da Atmosfera, Lisbon, Portugal
[5]European Centre for Medium-Range Weather Forecasts, reading, UK
[6]Laboratoire de Météorologie Dynamique/IPSL/Sorbonne Université/CNRS, UMR 8539, Paris, France
[7]Atmospheric Numerical Weather Prediction Research Section, Environment and Climate Change Canada, Dorval, Quebec, Canada
[8]Direction des Systèmes d'Observation, Météo-France, Toulouse, France

*Correspondence to*: Patrick Le Moigne (patrick.lemoigne@meteo.fr)

**Abstract.** The Antarctic Plateau, characterized by cold and dry weather conditions with very little precipitation, is mostly covered by snow at the surface. This paper describes an intercomparison of snow models, of varying complexity, used for numerical weather prediction or academic research. The results of offline numerical simulations, carried out during 15 days in 2009, show that the simplest models are able to reproduce the surface temperature as well as the most complex models provided that their surface parameters are well chosen. Furthermore, it is shown that the diversity of the surface parameters of the models
strongly impacts the numerical simulations, in particular the temporal variability of the surface temperature and the components of the surface energy balance. The models tend to overestimate the surface temperature by 2-5 K at night and underestimate it by 2 K during the day. The observed and simulated turbulent latent heat fluxes are small, of the order of a few W m$^{-2}$, with a tendency to underestimate, while the sensible heat fluxes are in general too intense at night as well as during the day. Finally, it is shown that the most complex multi-layer models are able to reproduce well the propagation of the daily
diurnal wave, and that the snow temperature profiles in the snowpack are very close to the measurements carried out on site.

## 1 Introduction

Snow is an essential component of the Earth's climate system. It plays a major role in climate regulation, as a water resource and as a key element of the landscape, for human societies and natural environments. It is known that snow cover has a profound effect on the Earth's surface, mainly by modifying the surface albedo, roughness and by thermally insulating the
underlying ground from the atmosphere. Furthermore, snow cover varies considerably in time and space and modulates radiative fluxes and fluxes of heat, momentum, and moisture between the surface and the atmosphere. Heat exchange between the atmosphere and the surface occurs through non-radiative fluxes, namely latent and sensible heat fluxes. In Antarctica, and



especially in the interior of the continent, as at Dome Charlie (Dome C hereafter), the small amount of available energy and the very cold temperatures that make the air dry and the specific humidity low induce very low latent heat fluxes (sublimation or solid condensation). Because of the strong reflection of incident solar radiation and heat loss through thermal radiation emission, the surface of the snowpack is generally colder than the atmosphere (Van den Broeke et al., 2005). In this case, it is the atmosphere that supplies energy to the snow surface. The insulating character of snow plays an important role in the surface-atmosphere coupling in snow-covered regions, either in areas temporarily covered with snow by precipitation events, in the plains or in the mountains, or in regions covered with snow throughout the year, such as the ice caps of the polar regions. At high altitudes and in the polar regions, the snow cover accumulates to form firns and turns into ice. For these reasons, the modelling of snow under these conditions is very important for climate. Furthermore, the improvement of snow processes in numerical weather prediction and climate models has always been an important area of research because of the challenges they represent.

Over time, various snow model intercomparison exercises have been carried out. These have allowed the comparison of snow models and even specific parameterizations of these models in order to better understand the processes studied and, if necessary, to improve them. Some studies have compared energy and mass balances, but for a limited number of snow models. For example, Essery et al. (1999) compared four snow models for a French Alpine site and found that the results were satisfactory on average, although there was considerable variability in the ability of the models to simulate the snow water equivalent, mainly due to the varying complexity of the models involved. They showed that the models were able to simulate comparable snow durations but that the peak snow accumulation and melt runoff were very different. Fierz et al. (2003) studied the energy balance of four snow models at a site in the Swiss Alps. They highlighted the importance of properly representing surface characteristics such as albedo (impact on radiation fluxes) and roughness length (turbulent fluxes) as well as heat conduction and water phase change processes within the snowpack. In a study comparing a simple and a more complex model, Gustafsson et al. (2001) found that the uncertainty in the surface parameters was more important than the model formulation. Jin et al. (1999) compared three snow models of varying complexity in three general circulation models (GCMs hereafter) and showed good agreement in surface flux, temperature and snow water equivalent of the models on a seasonal scale but poorer agreement on a diurnal scale for the simplest model, due to the failure to represent the water retention process within the snowpack. Boone and Etchevers (2001) also compared three models of varying complexity, but coupled to the same vegetation model. They showed the importance of surface parameters and the high variability in simulating the snow water equivalent. Koivusalo and Heikinheimo (1999) and Pedersen and Winther (2005) also showed the major role of surface parameters and the impact of the physics of the models on their ability to reproduce the surface energy balance.

Only a limited number of intercomparison exercises in which snowpack variables were explicitly considered have been undertaken. Thus, we can note the initiative of the World Meteorological Organization (1986), started in 1976, which compared eleven operational models in terms of snowmelt runoff on a varied data set and showed a good general behavior of the models



but already a certain variability linked to the diversity of the models that participated, each one having its own specificities according to the applications for which they were developed. Similarly, Schlosser et al. (2000) compared the simulation results of 21 models that represented the full range of complexity of snow patterns for a cold continental region in Russia. This study

was conducted as part of the Project for Interlaboratory Comparison of Land Surface Parameterization Schemes (PILPS, Henderson-Sellers et al. 1995). They found that there is considerable model variability for snow simulations, particularly with respect to snow ablation, which is of critical importance for predicted atmospheric fluxes and the hydrological cycle. General circulation model intercomparison studies of Atmospheric Model Intercomparison Project (AMIP) type, i.e. with climatologically imposed sea surface temperature and sea ice cover, have been conducted to evaluate continental-scale

estimates of snow cover and mass. In the AMIP1 (Frei and Robinson, 1998) experiment, comparisons were made of the representation of snow cover in the 27 GCMs that participated. In general, they found that seasonal variability was well represented by all models but that simulated interannual variability was underestimated. AMIP2 (Frei et al., 2005) focused on the ability of the 18 participating GCMs to simulate observed spatial and temporal variability in snow mass or snow water equivalent. Most models represented the seasonality of snow water equivalent and its spatial distribution reasonably well,

however, a tendency to overestimate the snow ablation rate was identified. Three others international intercomparison projects (PILPS2d, Slater et al. 2001; PILPS2e, Bowling et al. 2003; Boone et al., 2004) have focused on evaluating snowpack and runoff simulations for snow-influenced watersheds. In PILPS2d, the 21 surface schemes involved all showed roughly the same deficiency of too early snowpack melt. Boone et al. 2004 focused on the comparison of the water balance and in particular the daily snow depth over the Rhône catchment in France. One result was that models that explicitly represented the physics of

the snowpack performed better than the simplest models. In addition to the comparisons of surface schemes used in atmospheric models, other exercises more specifically dedicated to the study of processes in the presence of snow have been conducted. In the first phase of SnowMIP (Etchevers et al., 2004) comparisons of simulations of the surface energy balance and the snow water equivalent over a snow-covered low vegetation site were made. It was shown that model complexity played a dominant role in simulating the net infrared radiation budget. The same type of study was then conducted over forest areas

and results from the SnowMIP2 experiment (Essery et al., 2009) showed that many land surface models represent a sufficient range of processes that can be calibrated to well reproduce the mass balance of forest snowpack while simultaneously providing reasonable estimates of albedo and canopy temperatures that are essential for simulating the surface energy balance. More recently, an intercomparison of current ESM models has been conducted (Krinner et al., 2018) in an attempt to systematically integrate into future Coupled Model Intercomparison Project (CMIP)-type exercises an evaluation of snow models in order to

improve them. They showed that there is a large dispersion in the complexity of the snow schemes, thus pointing to the interest in improving the simplest as well as the most advanced parameterizations.

Within the framework of GABLS (Global Energy and Water Exchanges (GEWEX) Atmospheric Boundary Layer Study), intercomparison studies are conducted for boundary layer parameterization schemes used by numerical weather and climate

forecast models. For stable stratifications, the models still have significant biases, which depend on the boundary layer and



surface parameterizations used (Holtslag et al., 2013). The first three comparative GABLS studies (Cuxart et al., 2006; Svensson et al. 2011; Bosveld et al. 2014) only dealt with moderately stable conditions.

In GABLS4 (Bazile et al. 2014), the objective is to study the interaction of a high-stability boundary layer with a low-conductivity snow-covered surface with high cooling potential. In this context, an intercomparison exercise of snow models

forced by observations on the Antarctic plateau at Dome C has been carried out. This comparison complements coupled one-dimensional surface-atmosphere simulations and Large Eddy Simulations (Couvreux et al., 2020). Indeed, the day of December 11, 2009 was chosen as the reference day ("golden day") for the coupled simulations because it presented favorable conditions with low large-scale advection. The surface and snowpack model variables in these coupled simulations were initialized with an offline simulation having the same characteristics as in the coupled models.

The present study aims to evaluate the ability of the participating snow models to simulate the surface temperature, and even the temperatures in the snowpack for the more sophisticated models, as well as to evaluate the ability of the models to represent the surface energy balance at Dome C, i.e. under rather extreme cold conditions. This is quite a challenging exercise for models that have essentially been developed and validated at mid-latitudes and not necessarily exhaustively at the poles. These models are used in meteorological centers of numerical weather forecasting or laboratories that study the climate. The time period

covers a couple of weeks in December 2009 and the simulations are made in a standalone mode guided by the observations available on site. Models of varying complexity participate in this comparison and they use different surface parameters that have a strong impact on the simulations in this region. The scientific objectives addressed in this paper are:

- To briefly present the snow model intercomparison and position the GABLS4 experiment in relation to these snow-model intercomparison exercises;

- To study the variability of the simulations in surface temperature and more generally in surface energy balance;

- To show whether the simplest models can correctly simulate the surface temperature at Dome C, at least as well as the more complex models with an adapted set of parameters;

- To show the inter-model variability of the surface parameters used and the sensitivity of the models to these parameters;

- To show whether the most advanced multi-layer models simulate well the thermal stratification in the snowpack.

Section 2 describes the data used to generate the atmospheric forcing and the observed surface and snow data. It also provides a description of the participating models and the simulation protocols. In section 3, the results of all the simulations are presented. Finally, section 4 discusses the results and draws conclusions from the study.

## 2 Data and methods

### 2.1 Models

10 snow models of varying complexity from seven weather and climate centers participated in this comparison. The varying complexity of the models lies in their ability to represent complex physical processes. For example, the multilayer models account for snow compaction, heat diffusion between layers, percolation of liquid water within the snowpack as well as the



possibility that that water may freeze. But at Dome C, it must be stressed that the temperature is always below freezing and there is no significant precipitation during this experiment. So thermal diffusion and snow-atmosphere interaction are the parts of the snow schemes that are evaluated. In contrast, single layer models have a simplified representation of the processes and therefore a limited number of prognostic variables, such as albedo or snow density. The single-layer models involved are the Global Deterministic Prediction System version 4 (GDPS4 hereafter, McTaggart-Cowan et al., 2019a) from the Canadian Meteorological Center (CMC), D95 (Douville, et al., 1995) from the Centre National de Recherches Météorologiques (CNRM) and EBA (Bazile et al., 2002), also from the CNRM and which is a variant of D95 (in terms of albedo, thermal roughness length and snow melt calculations), the Carbon-Hydrology Tiled ECMWF Scheme for Surface Exchanges over Land CHTESSEL (Dutra el al., 2010; Boussetta et al., 2013) from the European Centre for Medium-Range Weather Forecasts (ECMWF) and NOAH (Mitchell 2005) from the National Center for Environmental Prediction (NCEP). Multilayer models are ISBA-Explicit-Snow (ISBAES hereafter, Boone et al., 1999; Decharme et al., 2016) and CROCUS (Brun et al., 1989; Vionnet et al., 2012) from CNRM, Community land Model version 4 (CLM4 hereafter, Oleson et al., 2010) from the Langley Research Center (LARC), LMDZ (Vignon et al., 2017b; Cheruy et al., 2020) from LMD (Laboratoire de Météorologie Dynamique) and IGE (Institue des Géosciences de l'Environnement), and lastly, JULES (Best et al., 2011) from the Met Office. Table 1 summarizes the organizations and models involved as well as the individuals who provided the results of the numerical simulations.

Table 1: List of models that participated in the GABLS4 snow model intercomparison at Dome C, Antarctica.

| Organization | Scientists running the models | Model name |
|---|---|---|
| Canadian Meteorological Center (CMC) | Ayrton Zadra | GDPS4 |
| Centre National de Recherches Météorologiques (CNRM) | Patrick Le Moigne | D95, EBA, ISBAES, CROCUS |
| European Center for Medium-range Weather Forecast (ECMWF) | Emanuel Dutra and Irina Sandu | CHTESSEL |
| IMSG@EMC/NCEP/NOAA | Anning Cheng | CLM4 |
| Laboratoire de Glaciologie et de Géophysique de l'Environnement (LGGE) | Etienne Vignon | LMDZ |
| Met Office | John M Edwards | JULES |
| IMSG@EMC/NCEP/NOAA | Weizhong Zheng | NOAH |

## 2.2 Simulation protocol

The models were run offline, i.e. guided by atmospheric forcing measured at Dome C, for a total simulation time of 15 days. Some surface parameters have been imposed in all models in order to reduce the dispersion of results and to be in the best conditions to perform an intercomparison. Thus, all participants were asked to carry out a reference simulation with an albedo





of 0.81, an emissivity of 0.98 (which corresponds to the hemispheric mean emissivity (Armstrong and Brun 2008)), dynamic and thermal roughness lengths of 0.001m and 0.0001m, respectively, and finally, for the single-layer schemes, to impose a snow density of 300 kg m⁻³. The albedo of snow depends on the zenith angle, but also on the grain size and cloud cover. At Dome C, as the sky is mostly clear, the effect of the solar zenith angle is prominent compared to the typical diurnal cycle. Warren (1982) showed that the albedo of snow was maximum when the sun was low while its effect was less when the sun

was at its zenith, it then allowed the surface to warm up, or at least to cool down less by radiative effect. Most models do not consider the variation of the albedo with the zenith angle, so a fixed average value is proposed in the experimental protocol, corresponding to the average value of the ratio between incident and reflected radiation measured at Dome C, over the considered period. Concerning the thermal emission of snow, the value of 0.98 is in the range of values commonly used for this type of medium. For the dynamic and thermal roughness lengths, values of 1 mm and 0.1 mm were chosen respectively.

The dynamic roughness length is close to that established by Vignon et al. (2017a) who studied the effect of sastrugi on flow and momentum fluxes and proposed using a thermal roughness length one order of magnitude smaller than the dynamic roughness length. This ratio of 10 is classically used in many models calculating fluxes at the surface-atmosphere interface. The density of snow on the surface can vary from 20 kg m⁻³ for fresh snow to 500 kg m⁻³ for old and wet snow. Measurements made at Dome C during the summer of 2014-2015 (Fréville, 2015) show that the snow density profile varies between 250 kg

m⁻³ and 310 kg m⁻³ between the surface and 20 cm depth (Gallet et al., 2011). A snow density of 300 kg m⁻³ was used for single-layer models in this study, whereas a snow density profile was prescribed for multi-layer models. The mid-layer depth (in meters) of each of the 19 layers was 0.0075, 0.0225, 0.04, 0.065, 0.1, 0.15, 0.22, 0.315, 0.445, 0.62, 0.87, 1.23, 1.83, 2.73, 3.73, 4.73, 7.23, 9.73, 10.29. The temperature and density snow profiles for December 1, 2009 at 00UTC were prepared from observed data. Each group was then free to provide the results of sensitivity tests that they deemed relevant.

In addition to the reference simulations, a rerun was proposed in order to better represent the diurnal cycle for the single-layer models, by imposing, in addition to the parameters listed above (fixed albedo, emissivity and dynamic to thermal roughness length ratio), the thermal coefficient of snow $c_s$ (K m² J⁻¹) which is directly involved in the equation for the evolution of the temperature along the vertical:

$$
\begin{cases}
C_{snow} \times \dfrac{\partial T(z,t)}{\partial t} = \dfrac{\partial}{\partial z}\left(\lambda(z)\dfrac{\partial T(z,t)}{\partial z}\right) & (1) \\
c_s = (h_{snow} \times C_{snow})^{-1} & (2)
\end{cases}
$$


Where $\lambda(z)$ is the heat conductivity of snow, $h_{snow}$ the snow depth (m) and $C_{snow}$ the volumetric heat capacity of the snow (J K⁻¹ m⁻³).

Moreover,

$$
C_{snow} = c_i \times \frac{\rho_{snow}}{\rho_i} \qquad (3)
$$



where $c_i$ and $\rho_i$ are the heat capacity and density of the ice respectively. Combining equations (2) and (3) gives finally equation (4):

$$c_s = \rho_i \times (h_{snow} \times \rho_{snow} \times c_i)^{-1} \qquad (4)$$

Taking a thickness of $h_{snow} = 5$ cm, densities of snow and ice of 300 kg m$^{-3}$ and 900 kg m$^{-3}$ respectively, and the heat capacity of ice $c_i = 1.895 \times 10^6$ J K$^{-1}$ m$^{-3}$ we obtain according to equation (4) the value of $c_s = 3.166 \times 10^{-5}$ J$^{-1}$ K m$^2$.

The rerun (XP1 below and corresponding to the experiments suffixed with "_new") was proposed to the modelers, essentially to try to limit the dispersion of the models in terms of surface temperature, but also to see whether or not this dispersion was reduced in the coupled single-column simulations (the latter is not addressed in this study). The drawback is that not all models were able to perform this new experiment, either due to lack of time or because the results were from an operational model that did not allow adjustment of certain parameters or variables in the schemes. Although not all of them participated, it is

interesting to study the impact of the proposed changes (XP1) compared to the previously obtained set of simulations (with or without calibration, XP0). We considered only the models that ran the initial simulations, with the desired sensitivity tests (XP0) and rerun (XP1) to calculate the daytime and nighttime biases for XP0 and XP1, as well as the RMSD difference between XP1 and XP0, to evaluate the impact on the model error.

## 2.3 Forcing data

Data describing the local climate were measured at Dome C on a mast equipped with sensors (Genthon et al., 2021), for a 15-day period from December 1 to December 15, 2009, the period during which air and surface temperatures are warmest in this region. They constitute a complete data set to feed surface models. The data collected on the mast at a height of 3.3 m are wind direction and speed with a Young anemometer, air temperature in a ventilated shelter with a PT100 probe, and specific humidity. In addition, air pressure was measured by a Vaisala sensor at a height of 1.2m and measurements of downwelling

infrared and visible radiation were made at a height of 3m with Kipp & Zonen sensors. Periods with missing data were filled with ERA-Interim reanalysis. Due to the lack of precipitation in the period, precipitation rates were set to zero. The above set of variables were averaged every 30 minutes to generate a continuous forcing over the study period. Table 2 describes the near-surface variables available generated from measurements made at the site and presents some metadata such as instrument type and measurement height.

Table 2: Description of instrumental devices.

| Variable | Sensor | Unit | Position | Height (m) |
|---|---|---|---|---|
| Wind direction | Young 05103 | degrees | Mast | 4.6 |
| Wind speed | Young 05103 | m s$^{-1}$ | Mast | 4.6 |
| Pressure | Vaisala RS92-SGP | Pa | Mast | 1.2 |
| Air temperature | PT100 or Vaisala HMP155 | K | Mast | 4.6 |



| Longwave incoming radiation | Kipp & Zonen CG4 | W m$^{-2}$ | Mast | 3 |
|---|---|---|---|---|
| Shortwave incoming radiation | Kipp & Zonen CM22 | W m$^{-2}$ | Mast | 3 |
| Specific humidity | Vaisala HMP155 | kg kg$^{-1}$ | Mast | 4.6 |

In situ measurements were available between 8 November 2009 and 1 January 2010 and have allowed to build an atmospheric forcing over a 15-day period starting on 1 December 2009 at 00UTC (i.e. 8LT). Figure 1 shows the temporal evolution of these variables, which constitute the meteorological forcing used for the offline simulations.





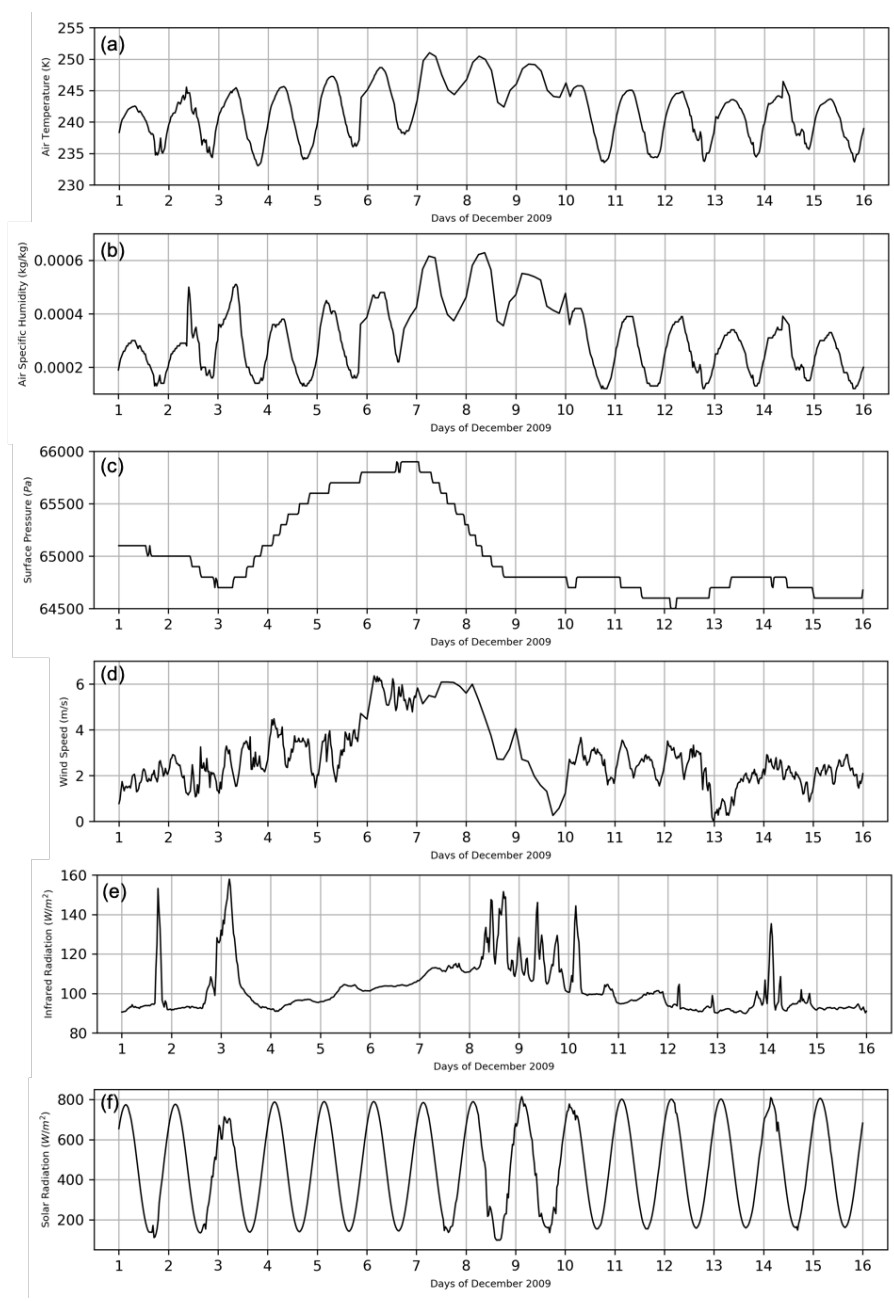


**Figure 1: Temporal evolution of: (a) air temperature, (b) air specific humidity, (c) surface pressure, (d) wind speed, (e) downward infrared radiation, and (f) direct solar radiation for the 15-day period of the offline simulations.**

Over the entire period, temperature, air humidity and wind speed data were missing for days 7, 8 and 9. The choice was to

replace them with data from the ERA-Interim reanalysis (Dee et al., 2011) for these three days and not to re-scale the measurements. For wind, the measurements showed good agreement with the reanalysis. The reanalysis tends to overestimate



the air temperature, especially at night with deviations of about 4 K while during the day this deviation is about 2 K. The low specific humidity is characteristic of a very dry air, and the difference between measurements and reanalysis is about 0.1 g kg⁻¹ during the day and night.


During the 15 days, the daily solar radiation varies relatively homogeneously and is characterized by an average diurnal amplitude that oscillates between 180 W m$^{-2}$ when the sun is low on the horizon and 800 W m$^{-2}$ when it is at the zenith. Infrared radiation shows a higher temporal variability with low values around 90 W m$^{-2}$ and higher values around 140 W m$^{-2}$, corresponding to cloudy periods, visible in particular at the beginning (days 1, 2 and 3) as well as in the middle (days 8, 9 and

10) and at the end of the period (day 14). The effect of clouds is also noticeable on the solar radiation time series. The period is also characterized by a strengthening of the surface wind, from 2 m s$^{-1}$ to 6 m s$^{-1}$, associated with an increase in atmospheric pressure (days 5 to 8). This dynamic effect leads to an increase in specific humidity, related to the arrival of clouds, and an increase in air temperature, probably related to increased mixing in the lower layers or advection effects and a limitation of atmospheric stability and thermal inversion at the surface.

**2.4 Evaluation data**

Surface and snowpack measurements were used to evaluate the models. Satellite measurements of surface temperature from Moderate-Resolution Imaging Spectroradiometer (MODIS) and the Infrared Atmospheric Sounding Interferometer (IASI) sensor complemented the continuous measurements from Baseline Surface Radiation Network (BSRN). Figure 2 shows the measurements from these sensors over the 15-day period.

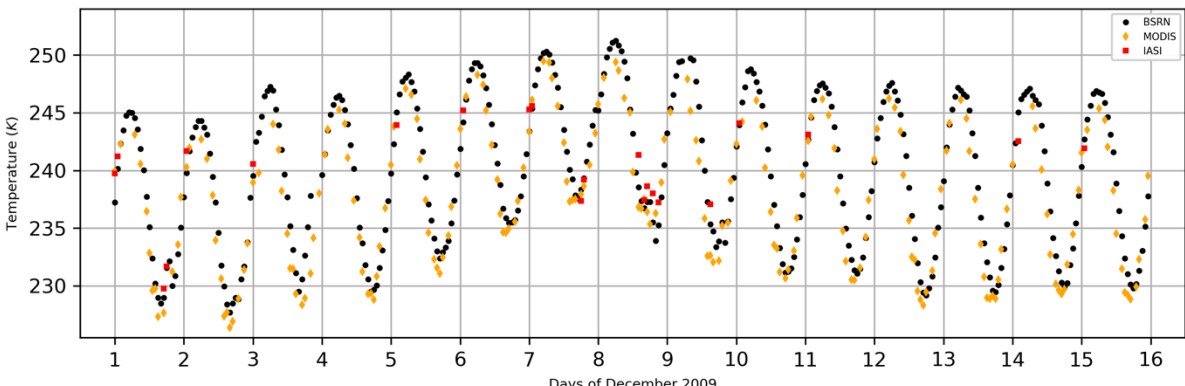


**Figure 2: BSRN, MODIS and IASI observation of surface temperature.**

In addition, measurements of the snow temperature profile, made by the Institute for Environmental Geosciences, allowed the characterization of the thermal structure of the snowpack and evaluation of the most sophisticated models with a multi-layer

vertical discretization (Brucker et al., 2011). The first temperature probe was installed at 10cm in the snow and the deepest at



21m. Over time snow was carried by the wind and accumulated on the measurement area. An annual accumulation of 8 cm per year is estimated at Dome C (Genthon et al., 2016, Picard et al., 2019), which corresponds in December 2009 to an accumulation of 23 cm of snow and therefore the first measurement in the snow corresponds to a depth of 33 cm. For the two weeks studied, a number of temperature measurements were missing in the snowpack. In particular, the period from December 250 7 to 11, 2009 was missing and the choice was made to fill it in to study the progression of the diurnal thermal wave in the snowpack over time and its representation in the multilayer models.

The gap-filling method is based on the simulation with the detailed multilayer model CROCUS, for which we consider that the temporal variability of the temperature in the snowpack is well simulated. Indeed, this model has already been evaluated 255 by Brun et al. (2011) over the Antarctic Plateau and had simulated the snowpack well. The CROCUS model configuration chosen in this study replicates that used by Brun et al. (2011). Details of the gap-filling method are presented in Appendix 1. Figure 3 shows the temperature of each snow layer to a depth of 423 cm (gap-filling is performed to a depth of 21 m in the snowpack, combining measurements (black) and data from CROCUS (orange)).

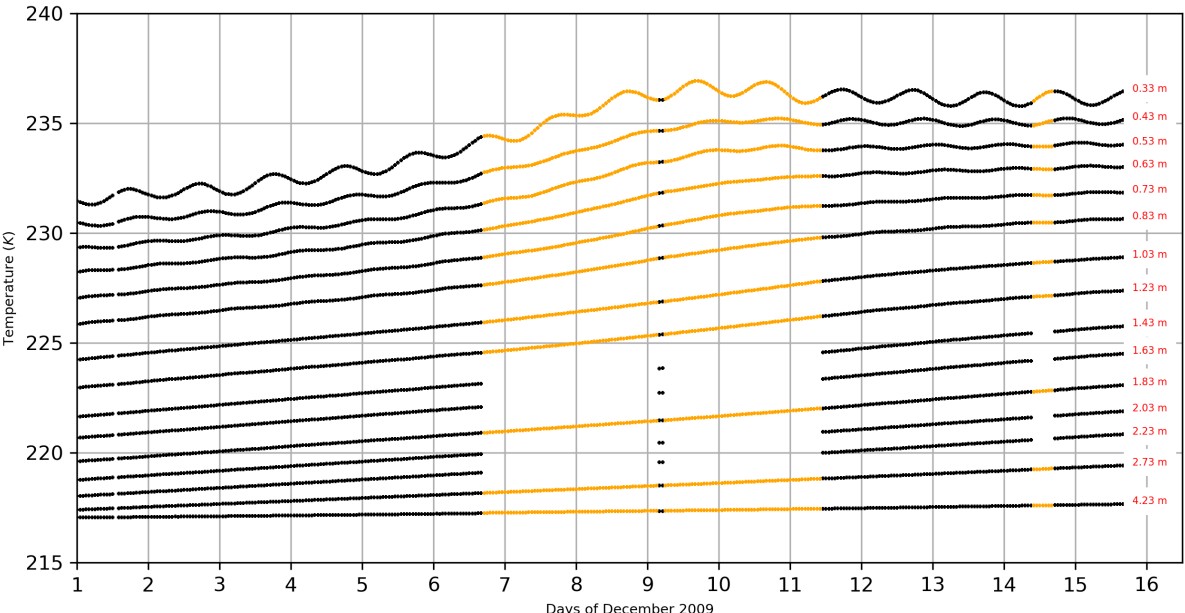

**Figure 3: Temperature measurements in the snowpack as a function of depth. The black dots represent the in situ measurements and the orange dots are the data reconstructed with the CROCUS model.**

Turbulent flux measurements by Eddy-Correlation are performed at high frequency (10 Hz) (Vignon et al., 2017b) at Dome C on an instrumented mast. The reconstruction of turbulent surface fluxes is a very complex exercise at Dome C, in particular 265 that of the latent heat flux of evaporation and sublimation, because the environmental conditions are extreme and the air is particularly dry. Scientists who have made measurements at Dome C have confirmed that comparisons of latent heat fluxes to

simulations are not completely relevant because of the large uncertainty in the measurement. However, we wanted to compare the simulated fluxes with the observations, even if the latter were questionable, because it was an additional way to characterize the variability of the simulations. At this time of the year, some convection is observed and during "daytime" (i.e. when the
sun is high above the horizon), although weak, the sensible heat fluxes are positive, that is, with the sign convention used, there is an energy transfer from the surface to the atmosphere. The sensible heat fluxes for the days of December 11, 12, 2009 and the following half-day were averaged at the hourly time step to be compared with the outputs of numerical simulations.

## 3 Evaluation of the modeled surface variables

### 3.1 Variability of surface parameters

This section aims to show the variability of the surface parameters of the different models, and how they evolve during the simulation, when they are not fixed, as is the case, for example, for the surface broadband albedo. As we will see, the choice of surface parameters is crucial to simulate the surface energy balance with sufficient accuracy. Table 3 gives for each model the values or ranges of variation of the surface parameters during the simulation. The albedos are close and represent well a reflective medium like snow. The albedo is a bit larger in GDPS4 and the snow surface will tend to reflect more solar radiation
during the day compared to the other constant albedo models. If we consider a radiative flux of 800 W m$^{-2}$ at the maximum of the day, a surface with an albedo of 0.83 leads to a net solar energy balance of 136 W m$^{-2}$ while it will be 160 W m$^{-2}$ for an albedo of 0.80. On the other hand, at night the minimum solar radiation is about 200 W m$^{-2}$ and the net balance will be 34 W m$^{-2}$ and 40 W m$^{-2}$ for albedos of 0.83 and 0.80 respectively.

Table 3: Range of variation of model surface parameters.

|  | Albedo | Emissivity | z0m (m) | z0m/z0h | Snow layers | Snow Density at surface (kg m$^{-3}$) |
|---|---|---|---|---|---|---|
| GDPS4 | 0.83 | 0.99 | 0.001 | 3 | 1 | 300 |
| D95 | 0.81 | 1.00 | 0.01 | 10 | 1 | 300 |
| EBA | 0.81 | 0.98 | 0.01 | 1 | 1 | 300 |
| ISBAES | [0.81,0.83] | 0.99 | 0.001 | 10 | 19 | [100,170] |
| CROCUS | [0.80,0.81] | 0.99 | 0.001 | 10 | 19 | [100,120] |
| CHTESSEL | 0.80 | 0.98 | 0.0013 | 10 | 1 | 300 |
| CLM4 | [0.84,0.88] | 0.97 | 0.0024 | 10 | 5 | 250 |
| LMDZ | 0.81 | 0.98 | 0.01 | 1 | 19 | - |
| JULES | [0.79,0.86] | 0.98 | 0.01 | 748 | 19 | [100,180] |
| NOAH | 0.81 | 1.00 | 0.01 | [1.6,6250] | 1 | 300 |




In contrast, Fig. 4 shows the modeled broadband albedos that vary over time. Indeed, the four models presented consider the variation of the albedo as a function of the age of the snow, which becomes denser under the effect of wind and compaction. Two of the models, JULES and CLM4, also consider the variation of albedo as a function of the zenith solar angle. We can see a great disparity in the albedos used. In particular, the daytime albedo of JULES (0.79) is lower than the others with a

consequence of a stronger warming of the surface. Overall, there is a decrease of about 1 % in the albedo value during the 15-day period. The ISBAES model has a larger albedo at the beginning of the simulation, it undergoes a more marked decrease between days 6 and 8. During this period, there is an increase in air temperature and humidity associated with an intensification in surface wind, which makes the snow denser on the surface. There is also a linear decrease in albedo, which is related to the nature of the model, which redistributes prognostic variables such as snow enthalpy to the snowpack thickness at each time

step. In general, the average thickness of the grains increases over time and decreases the albedo. The more significant decrease may be related, we believe, to a more pronounced increase in grain size due to the layer averaging effect. For the CROCUS model, there is a steady decrease in albedo over the period corresponding to the snow aging effect in connection with the steady increase in grain size and there is no impact of the wind intensification on albedo.

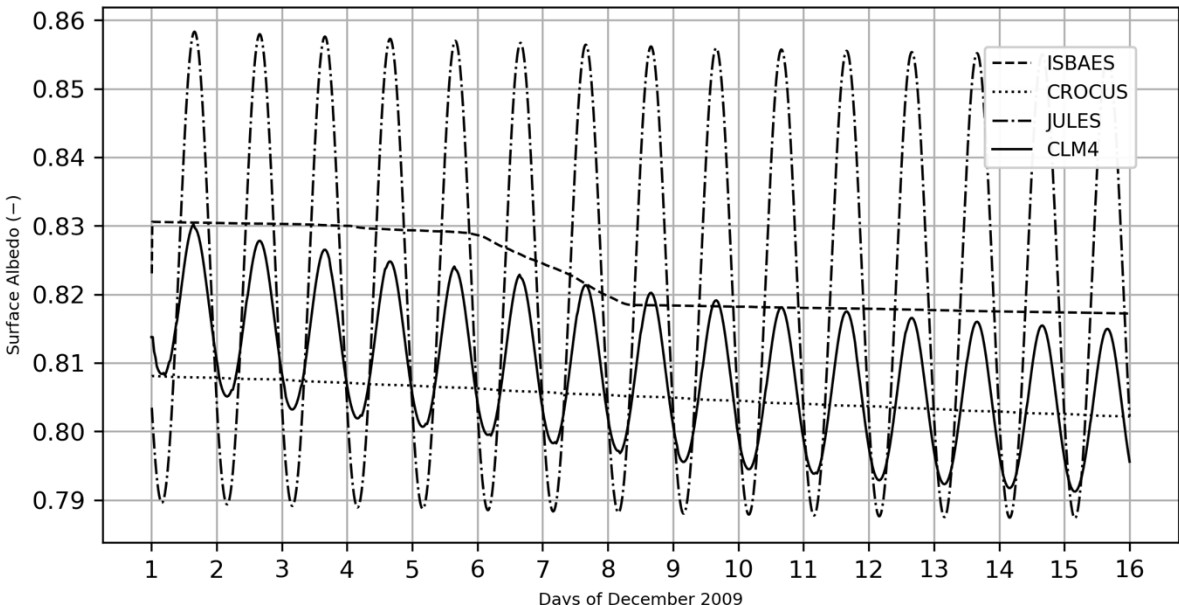

**Figure 4: Time evolution of surface albedo for ISBAES, CROCUS, JULES and CLM4 models.**

The second surface parameter playing an important role in the energy balance is the roughness length. Indeed, dynamic ($z0$) and thermal ($z0h$) roughness modulate the surface fluxes of momentum and sensible and latent heat. Vignon et al. (2017a) studied the variations of $z0$ from measurements at Dome C from which $z0$ was calculated using Monin-Obukhov (1954)

stability theory (MOST hereafter). They showed that the dynamic roughness varies between 0.01 mm and 6.3 mm for measurements made between January 2014 and February 2015 (average value of 0.56 mm) and that the value of $z0$ depends





on the wind direction: z0 is lower when the wind is aligned with the sastrugi, surface erosion patterns created by the wind. If it is difficult to estimate the dynamic surface roughness, the determination of the thermal roughness is also subject to many uncertainties (Andreas 2002) and most often the models use a thermal roughness proportional to the dynamic roughness. This

is the case for the models here except for NOAH whose z0h varies according to the stability of the air and ranges between $1.6 \times 10^{-6}$ m to $6 \times 10^{-3}$ m. The calculation of surface fluxes is based, for many models, on MOST which describes the influence of stability and roughness on turbulent exchange coefficients, the latter decreasing with increasing stability (Blyth et al., 1993). In Antarctica, turbulent flux exchange coefficients are low because the atmosphere is mostly stable and roughness is low (Deardorff, 1968). Surface roughness can also impact albedo by altering the effective zenith solar angle (Hudson et al., 2006)

and produce shadow zones at the surface (Leroux and Fily, 1998).

A snow density profile is provided for the multilayer models, with 100 kg m$^{-3}$ at the surface and 375 kg m$^{-3}$ at 10 m depth. Note that the single-layer models use a fixed density close to 300 kg m$^{-3}$ which corresponds to a depth of about 10 cm in the initial profile. The LMDZ model is a special case. Indeed, it is a ground thermal model with the thermal inertia of snow that is used and not really a snow scheme, which is why there is no snow density as such.

## 320  3.2 Surface temperature

The surface temperature directly influences the ambient air temperature and is itself directly influenced by the surface radiation budget. In summer, the diurnal cycle of the surface temperature is driven to the first order by the diurnal cycle of the solar radiation, which itself depends on the diurnal cycle of the solar zenith angle. At "night" (i.e. when the sun is low above the horizon), the zenith solar angle is low and the surface albedo is maximum. The infrared thermal radiation deficit then exceeds

the solar radiation gain and cools the surface. During the day, it is the opposite which occurs, the solar zenith angle is high while the albedo decreases inducing a heating of the surface by the solar radiation. The simulation of the surface temperature by the different models is a key point of our study. We were interested in the diurnal cycle of surface temperature over the 15 days of simulations, in particular the dispersion of all models but also their ability to simulate very cold diurnal cycles with strong thermal amplitudes. The simulations were compared to the available in situ and satellite measurements. Figure 5 shows

the time series of modelled surface temperatures (grey lines), on which the in situ measurements of the BSRN (black dots) and satellite measurements from MODIS (orange dots) and IASI (red dots) are also shown. Overall, the models are able to simulate the surface temperature quite well. However, there are strong disparities between some simulations, during both day and night, where the largest temperature differences can exceed 10K. All models overestimate the temperature at night on December 7 and 8, which correspond to missing data filled with ERA-Interim which is warmer than the locally observed temperatures.






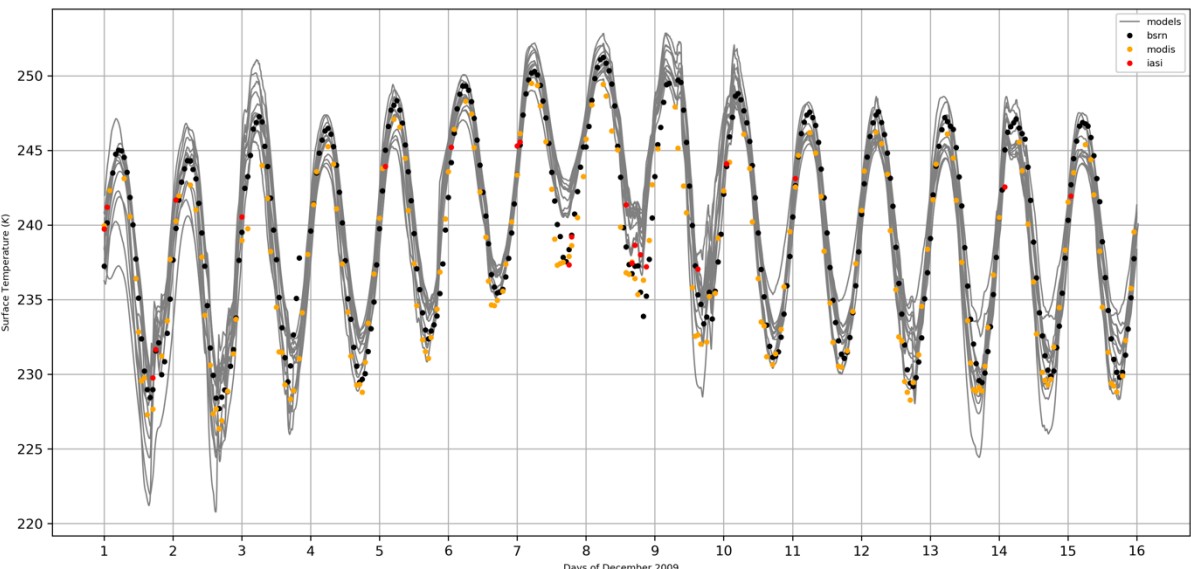

**Figure 5: Temporal evolution of the surface temperature observed by BSRN (black dots), MODIS (orange dots) and IASI (red dots) and simulated by the different models (grey lines).**

To better account for the behavior of the different models with respect to the observations, a probability distribution function (PDF) was computed for each model and for the BSRN observations and each PDF was fitted by a cubic function. MODIS and IASI observations were not used in this analysis because their number was insufficient for a robust statistical processing. In Fig. 6, the observed surface temperature PDF is indicated by the black dots, fitted by a cubic function (dashed line).

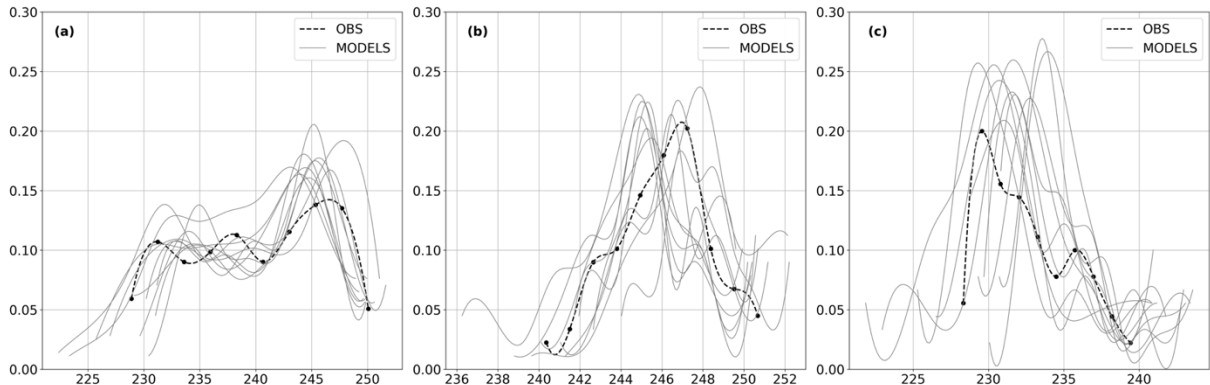


**Figure 6: Probability density of observed (dashed) and modelled (grey curves) surface temperature for: (a) all temperatures (left panel), (b) daytime (9LT-15LT) temperatures (middle panel) and (c) nighttime (21LT-3LT) temperatures (right panel).**



Over the whole period (Fig. 6.a), we notice a tri-modal distribution from observation and models that tend to underestimate
the maximum temperature and overestimate the minimum temperature. The decomposition into daytime (Fig. 6b) and
nighttime (Fig. 6c) time ensembles better illustrates these behaviors. Daytime is defined as the period 0UTC-6UTC (9LT-
15LT) while nighttime is defined as 12UTC-18UTC (21LT-3LT). In particular, during the day, most of the models have a
distribution fairly close to that of the observation but with a tendency to be about 2 K cooler. Some models have a distribution
closer to that of the observation. At night, two peaks appear which correspond to minimum temperatures (around 230 K) and
the second peak around 236 K which also corresponds to minimum temperatures but in warmer air between days 6 and 10.
The model distributions are more scattered at night. If the models manage to reproduce the nighttime cooling, many of them
tend to overestimate the surface temperature, from 2-3 K for some to 5 K for others. In Fig. 7 is shown the statistical behavior
of all the simulations performed, calculated at hourly intervals, in terms of bias and root-mean-square deviation (RMSD).
Indeed, each contributor was allowed to send the results of several realizations of the proposed simulation. On the x-axis of
this figure we find the name of an experiment, composed of the name of the model and a suffix corresponding to the test
performed. Note that the experiments with the extension "_new" correspond to the rerun which is described below.

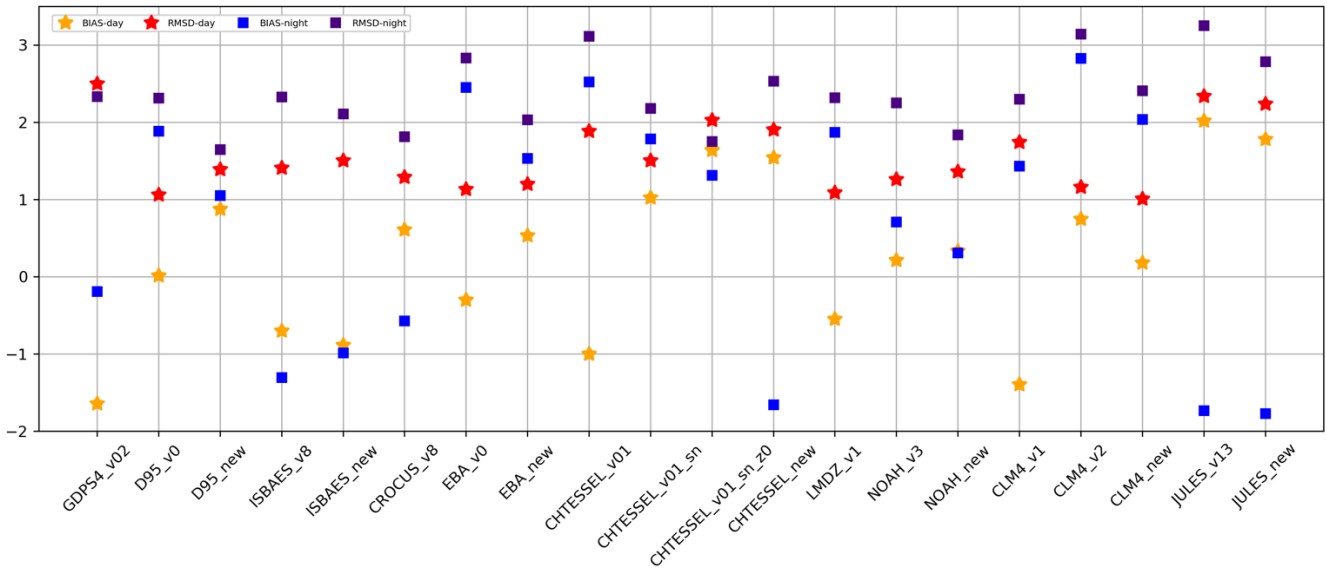

Figure 7: Statistical scores (BIAS and RMSD) during the day and night for the simulations performed for each model configuration.

### 3.3 Impact of the rerun on the surface temperature simulations

The conditions imposed for the rerun show that the daytime RMSD varies only slightly between XP0 and XP1 with sometimes
smaller errors for XP0 and other times for XP1 as shown in Table 4.

Table 4: Impact of rerun on BIAS and RMSD of model surface temperatures.

| | Bias (K) | RMSD (K) | RMSD (K) |
|---|---|---|---|
| | | | |





| Center | Model | Day | | Night | | Day | | Night | | XP1 - XP0 | |
|--------|-------|-----|-----|-------|-----|-----|-----|-------|-----|-----|-----|
| | | XP0 | XP1 | XP0 | XP1 | XP0 | XP1 | XP0 | XP1 | Day | Night |
| CNRM | D95 | 0.01 | 0.87 | 1.88 | 1.05 | 1.06 | 1.39 | 2.31 | 1.65 | 0.33 | -0.66 |
| CNRM | IES | -0.70 | -0.88 | -1.31 | -0.98 | 1.41 | 1.51 | 2.33 | 2.11 | 0.10 | -0.22 |
| CNRM | EBA | -0.30 | 0.53 | 2.45 | 1.53 | 1.13 | 1.19 | 2.83 | 2.03 | 0.06 | -0.80 |
| ECMWF | CHTESSEL | 1.63 | 1.54 | 1.31 | -1.66 | 2.02 | 1.91 | 1.75 | 2.53 | -0.11 | 0.78 |
| NCEP | NOAH | 0.21 | 0.33 | 0.71 | 0.31 | 1.26 | 1.36 | 2.25 | 1.84 | 0.10 | -0.41 |
| LARC | CLM4 | 0.75 | 0.18 | 2.82 | 2.03 | 1.16 | 1.01 | 3.14 | 2.41 | -0.15 | -0.73 |
| MO | JULES | 2.02 | 1.78 | -1.73 | -1.77 | 2.34 | 2.24 | 3.25 | 2.78 | -0.10 | -1.01 |

On the other hand, the RMSD is significantly improved at night for almost all models with improvements up to 1K. The
majority of the models have a smaller daytime bias than the nighttime bias for both XP0 and XP1, confirming the greater
difficulty of the schemes in representing the more stable conditions at night. This can be attributed to the snow scheme (in
particular albedo, emissivity, thermal coefficient of the snow and grain size) or to the parameterization of the turbulent fluxes
at the surface-atmosphere interface (dynamic and thermal roughness lengths involved in the calculation of the turbulent
exchange coefficients, as well as air stability criterion), in addition to the surface temperature itself depending on the albedo,
emissivity and thermal coefficient of the snow. Moreover, XP1 type experiments tend to show larger biases, especially during
the day but not for all models, and tend to decrease them at night. Therefore, in order to propose a comparison of all the models,
we decided to retain the best simulation of each model, performed in the XP0 framework. Each model is therefore evaluated
separately from the in situ observations, it is also a challenge of this intercomparison to learn from the different models and
see what could be improved. To do this, a comparison of the simulated and observed time series was carried out by separating
the night periods, i.e. corresponding to the hours between 12UTC and 18UTC from the day periods between 00UTC and
06UTC. This choice was motivated by the very strong diurnal amplitude at Dome C and the need to avoid error compensation
during bias calculations. Biases, root-mean-square error (RMSE), correlations were calculated on hourly data considering for
each observation the closest simulation time. The results obtained are summarized in Taylor diagrams presented in Fig. 8.

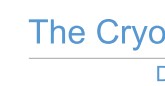
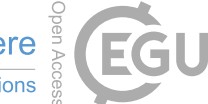


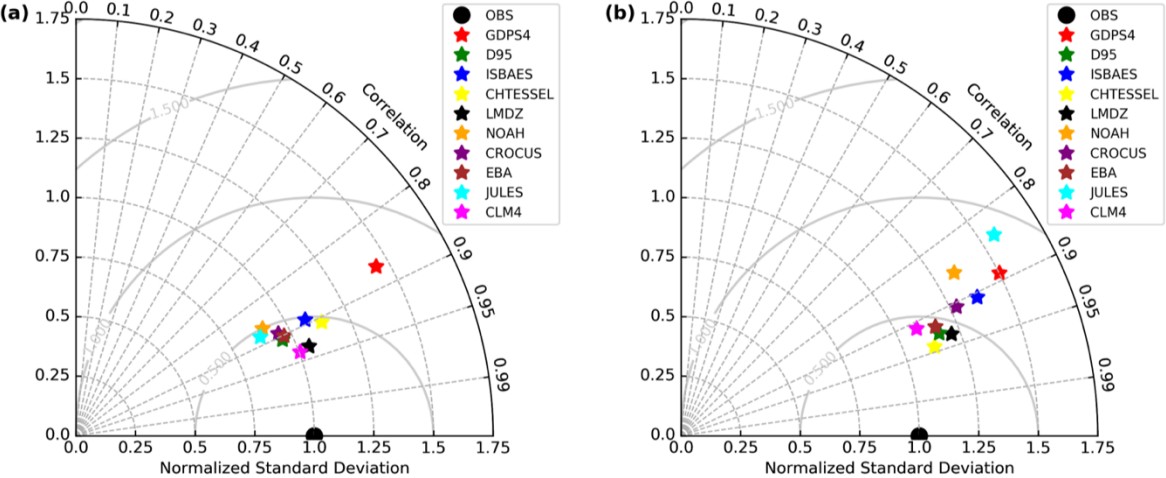

**Figure 8: Taylor diagram comparing the surface temperature scores of the different models (a) during the day and (b) at night.**

As a result, most of the models manage to represent the surface temperature during the day, except for the GDPS4 model which presents a higher error than the others. On the other hand, the results at night confirm the distributions of the PDFs, with a greater dispersion, a correlation that is fairly homogeneous and high around 0.9 and a root mean squared error that varies

from simple to double. It should be noted that the single layer models (D95, CHTESSEL, EBA) have sometimes better results than the more sophisticated models which have to represent more physical processes, such as the evolution of albedo with time, the increase of snow density by compaction, among others. The advantage of these simple models is that they are able to represent well the exchanges at the interface surface-atmosphere thanks to adapted surface parameters, such as the albedo and the heat transfer coefficient in the snow.

**3.4 Sensible and latent heat flux**

In this section, model comparisons to turbulent sensible and latent heat flux measurements, for the original versions of the models (XP0), are presented. The estimation of the contribution of sensible and latent heat fluxes to the surface energy balance is based, for all the models considered, on MOST, that describes in particular the influence of atmospheric stability and surface roughness on the variability of the exchange coefficient used for the calculation of the fluxes. Indeed, the sensible and latent

heat fluxes, expressed in their bulk form, are proportional to the modulus of the wind speed multiplied by the vertical gradient of temperature and specific humidity between the surface and the air respectively. The proportionality coefficient is the surface turbulent exchange coefficient. An increase in air stability induces a decrease in the exchange coefficient (Kondo, 1975; Blanc, 1985; Blyth et al., 1993). Thus, in Antarctica, the stable boundary layer and low surface roughness induce very low turbulent fluxes exchange coefficients (Deardorff, 1968).




Eddy-Covariance measurements were performed during the two months December 2009 and January 2010 at Dome C and have characterized the sensible and latent heat flux for two and a half consecutive days, with the first day, December 11, 2009 corresponding to the golden day as defined in the experimental protocol of the GABLS4 intercomparison exercise. Figure 9 is a scatter plot that compares $Q_h$, the hourly sensible heat flux simulated by the different models to the observations. First of all,

the graph shows two clearly distinct classes, corresponding on the one hand to the night with observed flux values between -2.5 W m$^{-2}$ and +2.5 W m$^{-2}$ and on the other hand to the day with observed values between 2.5 W m$^{-2}$ and 15 W m$^{-2}$. At night, turbulence is lower than during the day, partly because the wind modulus is lower, but also because the air density is higher and reduces the air vertical motion. Indeed, for the days considered, the minimum wind speed observed is about 2 m s$^{-1}$ at night and 3.5 m s$^{-1}$ during the day. Moreover, the radiation balance is negative at night, leading to a cooling of the surface

temperature, and positive during the day, thanks to the incident solar radiation that heats the snow. The simulated sensible heat fluxes show a bimodal behavior, with symmetrical and opposite values for day and night. During the day, the models simulate sensible heat fluxes between 5 W m$^{-2}$ and 40 W m$^{-2}$ and at night between -40 W m$^{-2}$ and -5 W m$^{-2}$.

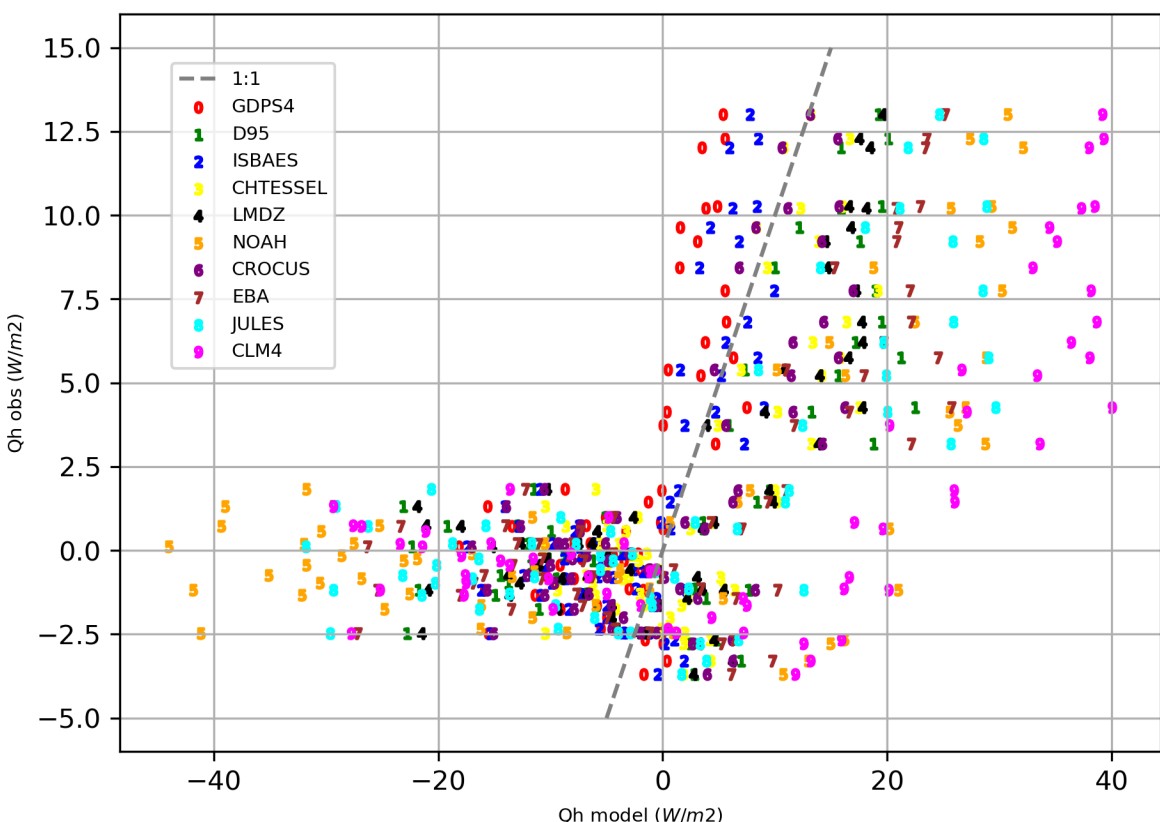

**Figure 9: scatterplot of sensible heat flux simulated (x-axis) and measured (y-axis). The grey dashed line represents the 1:1 line.**



In the same way, Fig. 10 is a scatter plot that allows us to compare $Q_{le}$, the hourly latent heat flux simulated by the different models with the observations. The first lesson that can be learned from this plot is that for all models except NOAH, the latent heat fluxes are lower than the values measured by Eddy-Covariance. Secondly, there is a separation around 5 W/m² for the observed $Q_{le}$ which corresponds to daytime for the higher values and nighttime and day/night transition for the lower values.

At night, the modeled values are low between -2 W m⁻² and +4 W m⁻² and during the day between 2 W m⁻² and 5 W m⁻² for most models except for CLM4 and NOAH which exhibit higher values.

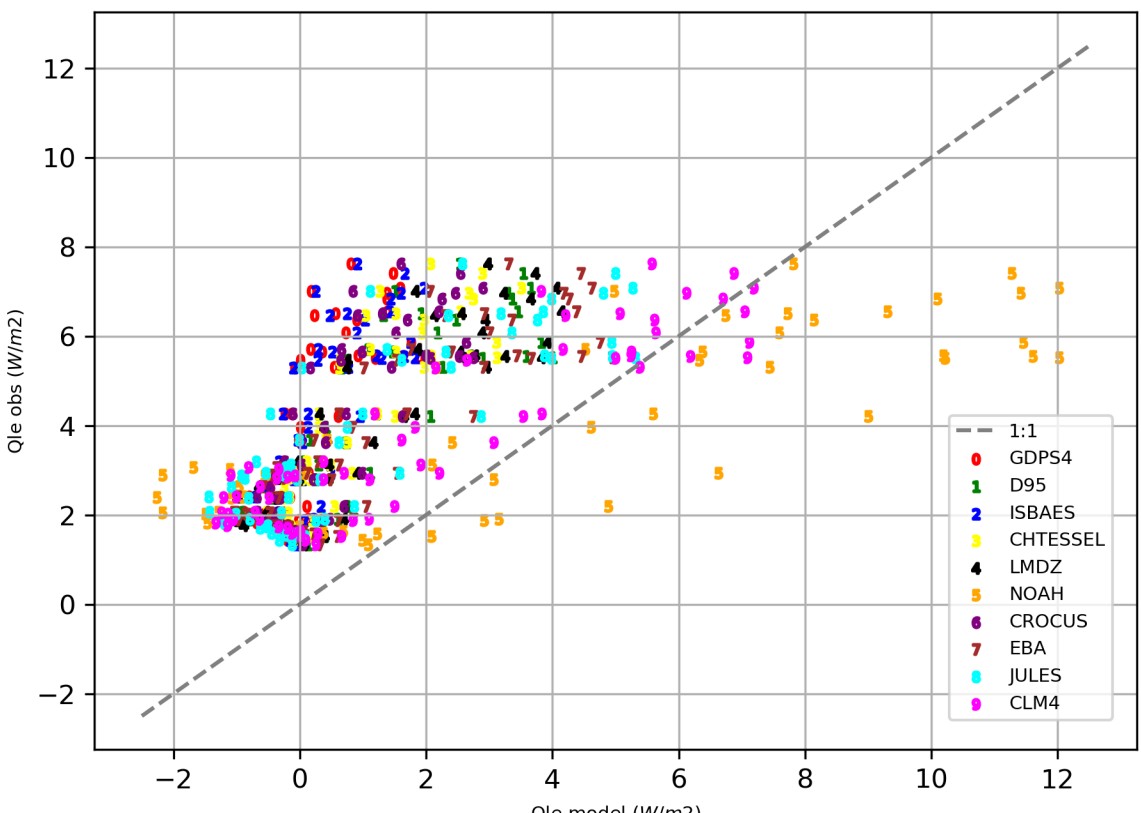

**Figure 10: scatterplot of latent heat flux simulated (x-axis) and measured (y-axis). The grey dashed line represents the 1:1 line.**

We are now interested in the variations of $Q_h$ for the different models. The sensible heat flux is written:

$$Q_h = \rho \times C_p \times \overline{w'\theta'} \qquad (5)$$

where $\rho$ is the air density, $C_p$ is the heat capacity at constant pressure and $\overline{w'\theta'}$ is the average correlation between the vertical velocity and potential temperature fluctuations. $Q_h$ is expressed in its Bulk form as follows:

$$Q_h = \rho \times C_p \times C_h \times U_a \times (T_s - T_a) \qquad (6)$$

where $C_h$ is the turbulent exchange coefficient, $U_a$ and $T_a$ are the wind speed and air temperature respectively, and $T_s$ is the temperature at the snow surface.





Each model solves its own energy balance and calculates in particular the surface temperature, a variable which is at the heart of the resolution of this balance. The variability of the models in terms of surface temperature will directly impact the variability

in terms of sensible heat flux. Similarly, the atmospheric conditions near the surface, i.e. temperature and wind speed, modulate the calculation of the $Q_h$ flux. Equation (6) also involves $C_h$ which depends on the dynamic and thermal roughness lengths as well as the stability of the air characterized by the gradient Richardson number $R_i$, except for the GDPS and CLM4 models. Figure 11 shows how $C_h$ varies as a function of $R_i$ for all models that provided values. We note a strong dispersion in the representation of the exchange coefficient $C_h$ with, depending on the model, four times higher values for instance in the case

of convection when $R_i$ is equals to -3. On the other hand, the values are very low for the stable atmosphere cases, i.e. when $R_i$ is positive, which is in good agreement with weaker turbulent exchanges or even almost zero in these conditions.

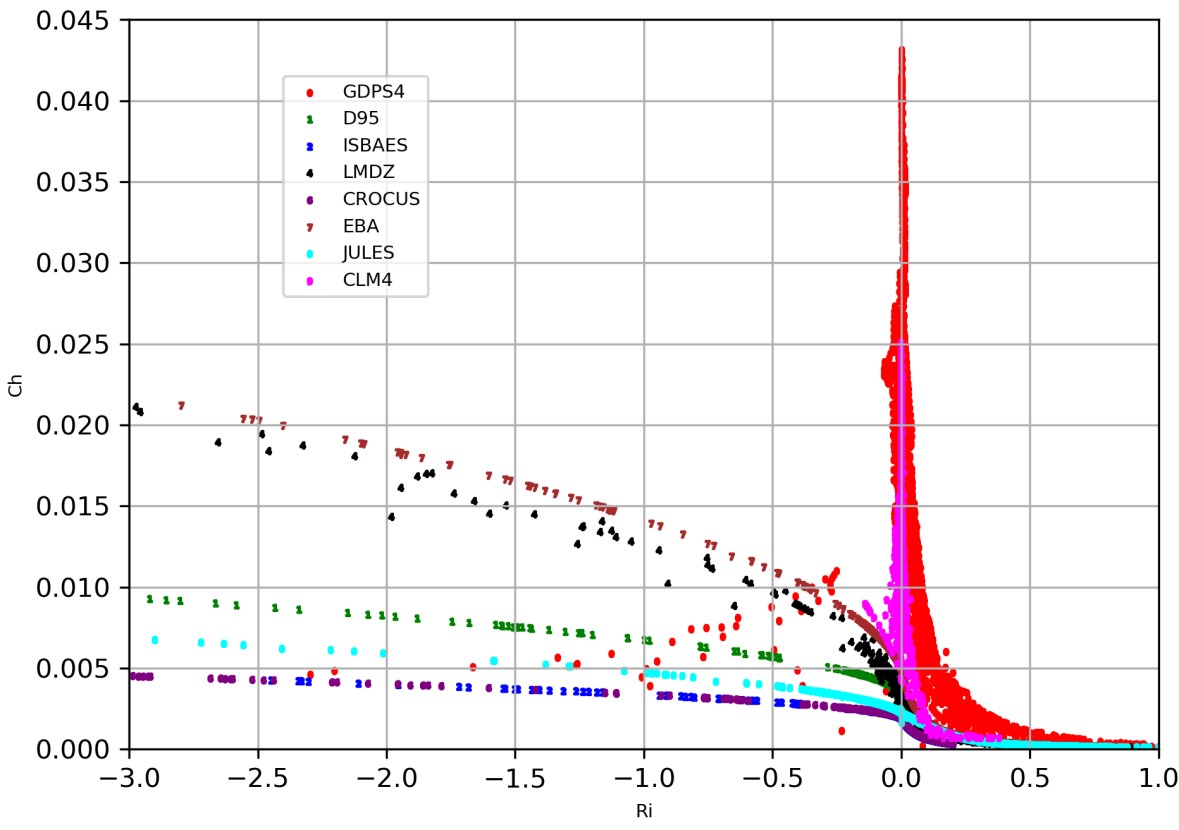

**Figure 11: Turbulent exchange coefficient for heat as a function of the Richardson number.**



### 3.5 Temperature profile in the snow

Some of the models are multi-layered and simulate the evolution of the profile of the variables that characterize the snowpack (density, temperature, enthalpy...). Thus, the JULES, ISBAES, CROCUS, LMDZ models have an identical vertical discretization of the snowpack in 19 layers as recommended by the experimental protocol while CLM4 has a discretization in 5 layers. Few observed data are available to make comparisons with the simulations, except for snow temperature. The snow temperature profiles of the multi-layer models were therefore evaluated over the 15-day period by comparison with measurements made at different depths. In order to make an identical comparison for all models, a vertical interpolation of the observed and simulated profiles was performed on a fine grid with a resolution of 1cm. The first statement concerns the results of CLM4, which are very different from the other models, with an unrealistic tendency to overheat the snow (the results are therefore not presented here). In Fig. 12 is shown the deviations of the temperature profiles from observations over time (the temporal evolution of the observed temperature profile is Fig. A2 in Appendix 1).

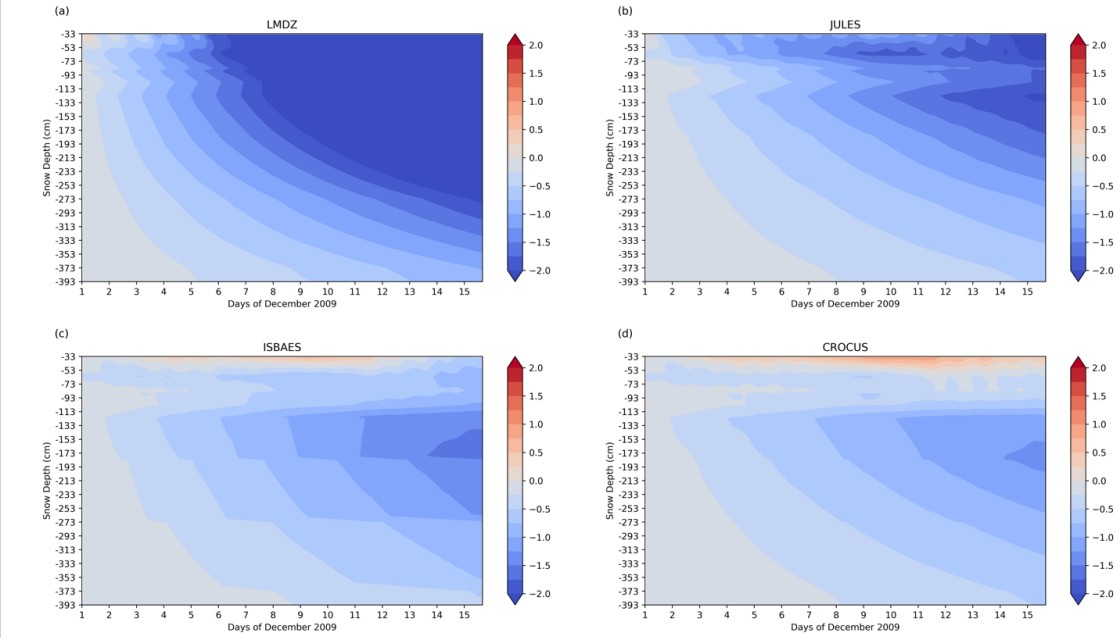

**Figure 12: Temporal evolution of deviations from observations of the temperature profile in snow.**

It can be seen that the vertical temperature profile is well initialized for the four models. The temporal evolution differs significantly from one model to another, except between ISBAES and CROCUS which have a large number of parameterizations in common. The LMDZ model tends to overcool the snowpack and this cooling appears at the surface and propagates into the deeper layers generating a generalized cold bias over the whole snowpack and reaching -2 K. The configuration of the LMDZ model for this intercomparison is particular since the model is not really a snow model but rather a soil model with the characteristics of snow. Sensitivity experiments on the coupled GABLS4 case revealed that the default



value of the snow thermal inertia set over Antarctica in LMDZ was way too high (close to a typical pure ice value). This parameter was therefore calibrated to a more realistic value after the GABLS4 exercise, leading to significant improvements of the temperature diurnal cycle (not shown here, see Vignon et al 2017b). For the remaining three models, similar behaviors can be observed for snow layers deeper than 1 m, but a different response between JULES and the two other models for the layers closest to the surface, between 33 cm and 1 m deep. Indeed, over time JULES tends to generate a cold bias reaching 2

K at the end of the period in the first meter, while ISBAES and CROCUS let heat penetrate more easily and the differences with observations vary between -0.5 K and +0.5 K. At the end of the period, CROCUS is the model with the lowest bias, of the order of -1 K at 153 cm, which is a very good score, while this bias is -1.5 K and -2 K for ISBAES and JULES respectively, indicating that these two models also perform well.

## 4 Concluding remarks

The study showed that the simple models performed well as long as the surface albedo and heat capacity were well prescribed. This is a very relevant finding for numerical weather prediction models because not all of them use very sophisticated snow models. Indeed, single-layer models are often preferred because multi-layer models represent a non-negligible cost in Numerical Weather Prediction (NWP) models (even if the cost of surface schemes is only a few percentage of the total model cost) and also because they significantly increase the complexity of the data assimilation schemes. However, multi-layer

models, which are more complex and have more advanced physics, can offer better performance. They are essential to study the internal dynamics of the snowpack and the penetration of the heat wave. One of the key variables for these models is the optical diameter of the snow used to characterize the snow microstructure which modulates the spectral albedo and has a direct impact on all snowpack processes, but unfortunately observations are rare and anyway difficult to use in an NWP context.

It was found that the intercomparison of snow models at Dome C was very valuable in several ways. First of all, the environmental conditions on the Antarctic plateau are extreme and testing the models under these conditions is very beneficial, especially for detecting their limitations. The results showed the good capacity of all models to represent correctly the temporal evolution of the surface temperature. The simplest as well as the most complex models are able to simulate the surface temperature thanks to a good simulation of the energy balance and all the better as the surface parameters are realistic. Indeed,

the models are very sensitive to surface parameters such as albedo and surface roughness and a large part of the inter-model variability comes from the disparity between these parameters in the models. Moreover, complex multi-layer models have shown their ability to represent not only the surface exchanges but also the thermodynamics of the snowpack. This aspect is very important when it comes to coupling these surface schemes with the atmosphere, as for example in climate models, which are used to study among others the impact of climate change on the snow cover and ice caps, with a particular attention to the

ice melting at the poles.



We chose to reconstruct the missing atmospheric forcing data using the ERA-Interim reanalysis data to avoid interpolation of the measurements, which would lead to uncertainty. The magnitude of the temperature difference between ERA-Interim and the measurements over the 15-day period reaches 4 K during the day and 2 K at night. This is a fairly large difference, which

was identified by Fréville et al. (2014), who found an overestimation of the turbulent mixing near the surface due to the parameterization of surface fluxes and a too large turbulent exchange coefficient. This was further investigated in Dutra et al. (2015), and the effective snow depth was even more guilty than the sensible heat flux.

However, snow has a low heat capacity and therefore the duration of the impact of such a difference was small. Other simulations (not shown) to study the impact of spinup on heat wave penetration also confirm this. And this is important because

the golden day selected for GABLS4 and the coupled surface-atmosphere simulations follows this period of missing data.

Surface flux comparisons are also subject to debate. Indeed, on the one hand, measurements by eddy-covariance present large uncertainties and on the other hand, the calculation by models, using the MOST theory, is not necessarily adapted to very stable conditions. Indeed, the surface parameterizations in stable cases have long been deficient and atmospheric models have

had difficulty in representing cases of high stability. For example, in this study, the turbulent exchange coefficient for heat is overestimated by all models compared to that diagnosed from observations (not shown). However, these measurements, even if they are subject to error, are invaluable in understanding the processes and in the possibility of comparing the results of the models with observations. Moreover, these observations are rather rare and having more measured and quality-controlled data would be a great progress. In the end, the temperatures simulated by these forced models are relatively good and an evaluation

of the models in coupled mode is the logical continuation of this work, which also requires good quality observation data sets.

**Appendix 1**

We consider the observed snow temperature profiles at two distinct times $t_1$ and $t_n$ and the open time interval $]t_1, t_n[$ during which the observations are missing. Moreover, for each snow layer, we know the temperatures simulated by CROCUS for each time $t_k$ ($k \in \{1, n\}$) of the interval $[t_1, t_n]$ and we calculate the temperature $T'_{OBS}(t_k)$ which would be observed at time

$t_k$ for a given layer if the temporal evolution of the temperature profile were that of CROCUS. We calculate the value $D(t_k)$ to be added or subtracted at time $t_k$ to the CROCUS temperature to find the observed value:

$$T'_{OBS}(t_k) = T_{CRO}(t_k) + D(t_k)$$

The sign prime indicates that it is an interpolated value and not the real observed value. For this, it is assumed that for each snow layer, $D(t_k)$ varies linearly between $D(t_1)$ and $D(t_n)$ which verify:

$$D(t_1) = T_{CRO}(t_1) - T_{OBS}(t_1)$$

and:




$$D(t_n) = T_{CRO}(t_n) - T_{OBS}(t_n)$$


where $T_{OBS}(t_1)$, $T_{OBS}(t_n)$, $T_{CRO}(t_1)$, $T_{CRO}(t_n)$ are the values of the temperatures observed and simulated by CROCUS at times $t_1$ and $t_n$ for the layer $j$ considered. It follows that:

$$T'_{OBS}(t_k, j) = T_{CRO}(t_k, j) + D(t_k, j)$$

Where:


$$D(t_k, j) = \left[ \begin{array}{l} (t_n - t_1) \times \left( T_{CRO}(t_1, j) - T_{OBS}(t_1, j) \right) \\ + (t_k - t_1) \times \left( T_{CRO}(t_n, j) - T_{OBS}(t_n, j) \right) \end{array} \right] / (t_n - t_1)$$

Figure A1 highlights the principle of the observed temperature reconstruction method.

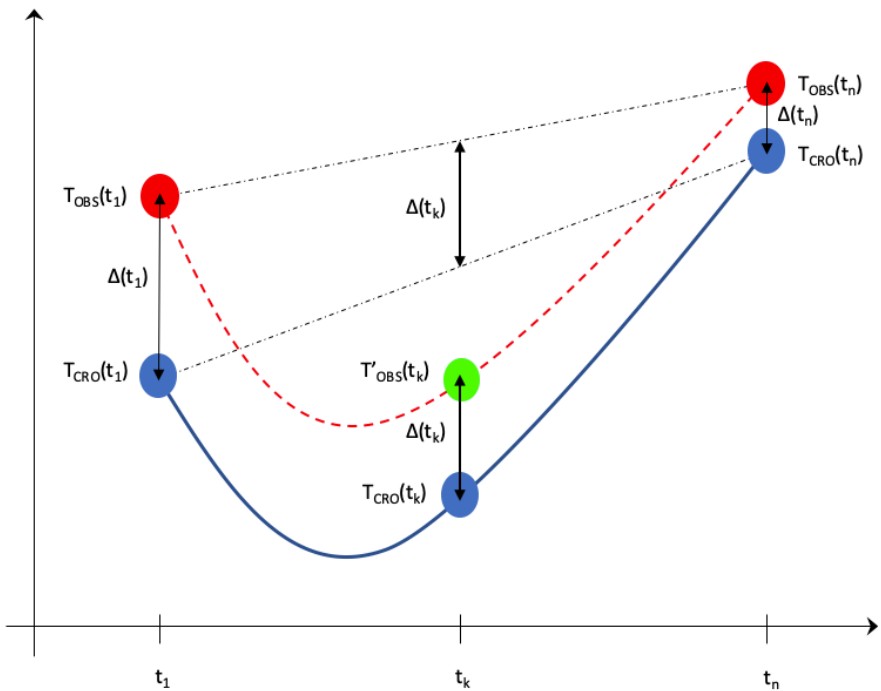

**Figure A1: Schematic diagram of the method of filling in the missing values observed from a numerical simulation with the CROCUS**
**model.**

In Fig. A2 is shown the temperature profile in the snowpack reconstructed from the measurements and completed by the temperatures interpolated by using the time variability of the simulated CROCUS temperatures for the different layers using the algorithm described above. This field was then interpolated on a 1 cm resolution vertical grid in order to make comparisons
with the detailed models which do not have the same vertical discretization.



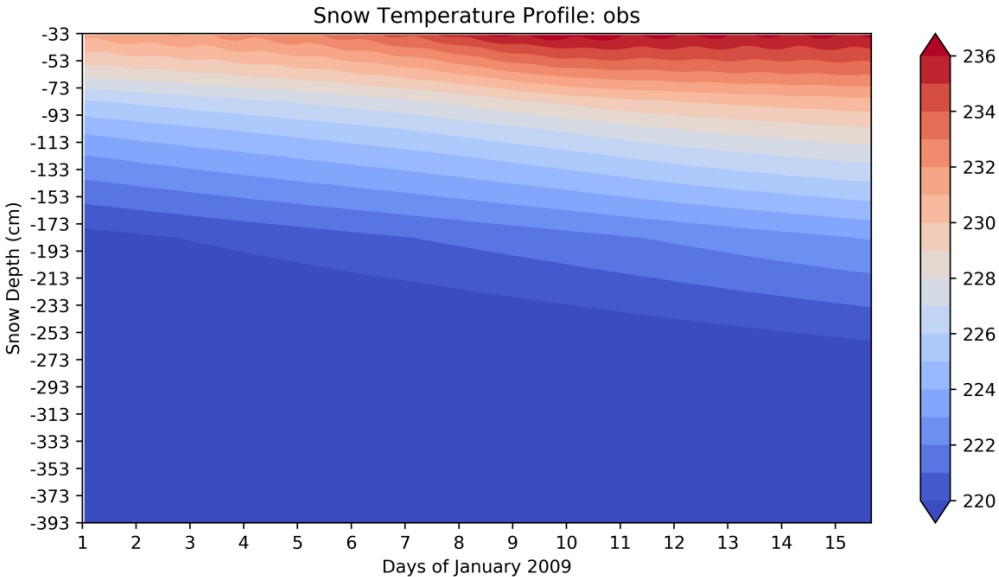

**Figure A2: Temperatures in snow as a function of time for the 15-day period in December 2009, interpolated to a fine vertical grid.**

**Data availability**

The data used in the figures, i.e., forcing data, simulations results, observations and the python scripts used to process the data, can be downloaded here: https://doi.org/10.5281/zenodo.5814726 (Le Moigne, 2022).

**Author contribution**

PLM and EB designed the experiments. AC, AZ, ED, EV, IS, JME, PLM and WZ carried them out. WM and OT provided the in situ measurements and surface fluxes. PLM prepared the manuscript with contributions from all co-authors.

**Competing interests**

The authors declare that they have no conflict of interest.

**Acknowledgments**

The first author would like to warmly thank E Brun for the support to this study and for the configuration of the Crocus model, and E Brun, A Boone and B Decharme for the constructive discussions on the snow models.



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
