# Peer review of "GABLS4 intercomparison of snow models at Dome C in Antarctica."

_The Cryosphere, 2022_

## Author Comment (AC1)

**RC1 (John King)**

**General**

Numerical Weather Prediction (NWP) models, atmospheric reanalyses and Earth System Models (ESMs) are being used increasingly to study the weather and climate of the polar regions. However, the land surface and surface exchange schemes used in such models have generally been optimised for performance in mid-latitudes and, while they may include a representation of snowpack processes, the snow models used may not correctly represent processes at work in the cold, dry and persistent snowpacks of the polar regions. This interesting and well-written paper reports an intercomparison of stand-alone snowpack models using forcing and validation data from Dome C, Antarctica, a location which is representative of the high Antarctic plateau. The intercomparison, carried out as part of the GEWEX GABLS project, included snow models that are components of a number of leading ESMs and NWP models and the results will be of value to users of those models, particularly those using these models in the polar regions. The paper is (mostly) written very clearly and presents useful results. I recommend that it should be published in The Cryosphere after attention has been given to the points listed below.

**Major points**

1. Section 2.2: I found this section quite difficult to follow. In particular, I found the different ways in which the various runs are referred to - "reference simulation", "XP0", "XP1", "_new" – rather confusing. Please try to adopt a clear and consistent terminology. Are the parameters shown in table 3 those that were used in the "reference simulation" (XP0)? I was also quite confused about how XP1 differed from XP0 – are the changes only applicable to single-layer models? As this section is describes a central part of the methodology of the study it needs to be rewritten to improve its clarity. It might help to add additional tables to highlight the differences between the XP0 and XP1 runs. You don't say anything in this section about how the models were initialised (presumably using observed snow temperatures?) or whether any spin-up was undertaken to avoid initial transients.

   Thank you for this very useful remark. It is true after reading again the manuscript that it was more than unclear. Section 2.2 was modified substantially in order to make it clear. In a few words, XP0 stands for the simulations with model's default values for surface parameters, some models have performed calibration. Then XP1 corresponds to XP0 with imposed parameters for albedo, emissivity, roughness length, density and the thermal coefficient of snow. Some multi-layer model also ran XP1 (but they did not impose density nor thermal coefficient of snow). New tables were added to clarify the differences between XP0, calibrated XP0 and XP1. As now indicated in Section 2.2, models were initialized from the observed snow temperature profile and no spin-up was done.

   Here is the new content of section 2.2:

The models were run offline, i.e. guided by the atmospheric forcing measured at Dome C, for a total simulation time of 15 days. Snow temperature was initialized from in situ

measurements. First, each group had to provide the results obtained with the default settings of the surface parameters of their model. The models were run offline, i.e. guided by the atmospheric forcing measured at Dome C, for a total simulation time of 15 days. This set of simulations is called XP0, it includes one simulation for each of the 10 models and the name of an experiment is made up of the name of the model suffixed by "_xp0". From the simulations in the XP0 set, each participant could propose additional simulations. Only CHTESSEL and CLM4 performed calibration simulations, aiming at minimizing the root mean square error on the surface temperature. CLM4 calibrated the surface albedo and CHTESSEL calibrated the snowpack thickness and from this calibration then calibrated the dynamic and thermal roughness lengths. The names of the corresponding experiments are CLM4_cal1, CHTESSEL_cal1 and CHTESSEL_cal2 respectively. In addition, a rerun was proposed in order to better represent the diurnal cycle and to reduce the dispersion of the surface temperature results, but also to see if this dispersion was reduced or not in the coupled single-column simulations (this last point is not addressed in this study).

Snow albedo depends on zenith angle, but also on grain size and cloud cover. At Dome C, because the sky is generally clear, the effect of solar zenith angle is prominent compared to the typical diurnal cycle. Warren (1982) showed that the albedo of snow was maximum when the sun was low while its effect was less when the sun was at the zenith, it then enabled the surface to warm, or at least cool less by radiative effect. Most models do not consider the variation of albedo with the zenith angle, a fixed average value is proposed in the experimental protocol, corresponding to the average value of the ratio between the incident and reflected radiation measured at Dome C, over the period considered. Concerning the thermal emission of snow, the value of 0.98 is within the range of values commonly used for this type of medium. For the dynamic and thermal roughness lengths, values of 1 mm and 0.1 mm were chosen respectively. The dynamic roughness length is close to that established by Vignon et al. (2017a) who studied the effect of sastrugi on flow and momentum fluxes and proposed using a thermal roughness length that is one order of magnitude smaller than the dynamic roughness length. This ratio of 10 is classically used in many models calculating fluxes at the surface-atmosphere interface. Snow density at the surface can range from 20 kg m$^{-3}$ for fresh snow to 500 kg m$^{-3}$ for old, wet snow. Measurements at Dome C during the summer of 2014-2015 (Fréville, 2015) show that the snow density profile varies between 250 kg m$^{-3}$ and 310 kg m$^{-3}$ between the surface and 20 cm depth (Gallet et al., 2011).

Therefore, all participants were asked to run a new simulation with an albedo of 0.81, an emissivity of 0.98 (which corresponds to the average emissivity of the hemisphere (Armstrong and Brun 2008)), dynamic and thermal roughness lengths of 0.001 m and 0.0001 m, respectively, and for single-layer schemes, to impose a snow density of 300 kg m$^{-3}$, as well as the snow thermal coefficient $c_s$=3. 166 ×10$^{-5}$ (K m$^2$ J$^{-1}$). This coefficient is directly involved in the temperature evolution equation along the vertical:

$$C_{snow} \times \frac{\partial T(z,t)}{\partial t} = \frac{\partial}{\partial z}\left(\lambda(z)\frac{\partial T(z,t)}{\partial z}\right) \qquad (1)$$

$$c_s = (h_{snow} \times C_{snow})^{-1} \qquad (2)$$

Where $\lambda(z)$ is the heat conductivity of snow, $h_{snow}$ the snow depth (m) and $C_{snow}$ the volumetric heat capacity of the snow (J K$^{-1}$ m$^{-3}$).
Moreover,

$$C_{snow} = c_i \times \frac{\rho_{snow}}{\rho_i} \qquad\qquad (3)$$

where $c_i$ and $\rho_i$ are the heat capacity and density of the ice respectively. Combining equations (2) and (3) gives finally equation (4):

$$c_s = \rho_i \times (h_{snow} \times \rho_{snow} \times c_i)^{-1} \qquad\qquad (4)$$

Taking a thickness of $h_{snow} = 5$ cm, densities of snow and ice of 300 kg m$^{-3}$ and 900 kg m$^{-3}$ respectively, and the heat capacity of ice $c_i = 1.895 \times 10^6$ J K$^{-1}$ m$^{-3}$ we obtain according to equation (4) the value of $c_s$. This rerun is named XP1 and the name of the simulations that refer to it consists of the name of the model suffixed by "_xp1".

Not all models were able to perform this new experiment, either due to lack of time or because the results came from an operational model that did not allow for adjustment of certain parameters or variables in the schemes. Although not all of them participated, it is interesting to study the impact of the changes induced by the XP1 configuration on the simulations of the XP0 ensemble, considering, when they exist, the calibrations. We therefore calculated the daytime and nighttime biases, as well as the difference in RMSD between XP1 and XP0 (or the simulation calibrated from a simulation of the XP0 ensemble), and evaluated the impact on the model error.

2. Section 3.4: The measured values of Qh and Qle on figures 9 & 10 indicate a Bowen ratio, Qh/Qle of O(1), which seems remarkably low at such low temperatures (see discussion in King et al, 2006, doi:10.1029/2005JD006130). Measuring Qle at such low humidities is very challenging and I suspect that there are very large uncertainties in the measurements.

   This is completely true and it was part of the discussion section, a comment was added in section 3.4 to emphasize the fact that the Bowen ratio is very small and probably a result of large uncertainties in the measurements of both Qh and Qle, particularly an overestimation of Qle, and also made a reference to the mentioned paper.

   Text added in section 3.4:

Figure 9 and 10 are scatter plots that compare $Q_h$ and $Q_{le}$, the hourly sensible and latent heat flux, respectively, simulated by the different models to the observations. We see at first that the measured latent heat flux is abnormally high. Indeed, as shown by King et al. (2006), this flux can only be of the order of a few W/m2 at Dome C, and that the closure of the energy balance has a high uncertainty. Thus, the reconstruction of heat fluxes from Eddy-Covariance measurements is likely to be subject to error and comparisons made here should be taken with caution.

3. Section 3.4, fig. 11: Looking at this figure, it is apparent that most of the models specify Ch as a universal function of Ri, but GDPS4 and CLM4 seem not to do so. What other factors are used in the calculation of Ch in these models? Would the curves for the different models collapse onto a universal curve if, instead of plotting Ch, you plotted Ch/Ch(Ri=0) – i.e. the ratio of Ch to its value under neutral stratification (which should only depend on the roughness lengths)? It would also be interesting to plot Ch calculated from the Dome C observations on this figure. From my own experience, I would expect to see a lot of scatter if you plotted points for each

30 minute observation, but if you averaged these together in bins of Ri you might get a useful set of points for comparison with the models.

Good remark, most models use Louis model to calculate Ch as a function of Ri. There are 3 exceptions here GDPS4, CLM4 and NOAH (NOAH is not in fig11 since Ri was not provided), for which the calculation of Ch is iterative and based on Monin Obukhov's theory. I have proposed a new figure to replace fig11, where GDPS4, CLM4 and NOAH are removed since their Ch does not depend on Ri. Then I followed your advice to normalize by the Ch at neutrality. This is the figure 11a I have now included in the manuscript, to show that the curves do not collapse into a single universal one, and which highlights the fact that most models have tuned their stability function from the universal one. I have calculated Ch from the observations and tried to compute the bulk Ri from the observation as well. ChObs is very small as compared to the modeled values. There is a large uncertainty in the calculation of Ri (z0_obs, z0h_obs, …) therefore I have not included it. But I have proposed another plot (Fig. 11b) which is the time evolution of Ch (in that way all models can be plotted) in the time window where I have observations (2.5 days) in order to show how different modeled Ch are from estimated ChObs.

The description of Figure 11 results has been modified:

Equation (6) also involves $C_h$ which depends on the dynamic and thermal roughness lengths as well as the stability of the air characterized by the gradient Richardson number $R_i$, except for the GDPS, CLM4, and NOAH models, for which the calculation of $C_h$ is iterative and based on Monin Obukhov's theory. Figure 11a shows how $C_h$ normalized by its value at neutrality (i.e. when $R_i=0$) varies as a function of $R_i$ for all models that provided values. that the curves do not collapse into a single universal one, and which highlights the fact that most models have tuned their stability function from the universal one. We note a strong dispersion in the representation of the normalized exchange coefficient with, depending on the model, values twice as large, for instance in the case of convection when $R_i$ is equals to -3. On the other hand, the values are very low for the stable atmosphere cases, i.e. when $R_i$ is positive, which is in good agreement with weaker turbulent exchanges or even almost zero in these conditions. To highlight the disparities in the Ch coefficient, the temporal evolution of Ch has been plotted in Figure 11b for all models, as well as the value of this coefficient calculated from the observations. Figure 11b shows that $C_h$ is simulated rather well for low turbulence conditions (low $C_h$) but is overestimated for the GDPS4, CLM4 and NOAH models. On the other hand, when the turbulence increases (December 13), these models simulate $C_h$ quite well. However, the variability of the simulated $C_h$ is then much greater.

4.  Section 4, concluding remarks. This study has focussed largely on snow models that are used within global models and have not been specifically optimised for polar conditions. It might be worth mentioning here work that has been done on developing polar-optimised snow/firn models for use within regional climate/NWP models, such as Polar WRF (Hines and Bromwich, 2008, 10.1175/2007MWR2112.1), MAR (e.g. Agosta et al, 2019, 10.5194/tc-13-281-2019) and RACMO2 (e.g. van Wessem et al, 2018, 10.5194/tc-12-1479-2018).

I agree that this is missing and I have added a comment to correct that:

This study has largely focused on snow models that are used within global models and have not been specifically optimized for polar conditions. However, it is important to note that work has been done to develop snow/firn models optimized for polar conditions for use in regional NWP and climate models, such as Polar WRF (Hines and Bromwich, 2008), MAR (Agosta et al., 2019), and RACMO2 (van Wessem et al., 2018).

**Minor points and typographical corrections**

Line 40: "firn", not "firns"

done

Line 146: "UK Met Office"

done

Table 1: Maybe add a column indicating which NWP/ESM/reanalysis models use the snow model that is being tested. I assume that "LMDZ" refers to the snow submodel used within the LMDZ global atmosphere model – doesn't the submodel have its own name?

Table 1 was removed (see RC2 comments)

Lines 171-174: Maybe include a table that gives the snow layer depths, densities, etc.?

This information has been added to the end of the new section 2.2 and a table was added:

For the multilayer models, the snow density and temperature profiles were initialized from observations. Note that the single-layer models use a fixed density close to 300 kg m$^{-3}$ which corresponds to a depth of about 10 cm in the initial profile, and their initial temperature was also provided from in situ measurements. Table 1 describes the vertical discretization and gives the initial temperature and snow density profiles. The LMDZ model is a special case. Indeed, it is a ground thermal model with the thermal inertia of snow that is used and not really a snow scheme, which is why there is no snow density as such.

Line 211: "…enabled us…" instead of "…allowed…"?

done

Figure 1: Pressure seems to have been recorded with only 1 hPa resolution? "Direct solar radiation…" in the caption should (I think) be "downward solar radiation"."Direct" would usually mean direct solar beam only, i.e. not including the diffuse component.

Looking at the pressure time series, it has not always been recorded with a 1hPa resolution but most of time it is the case. The manuscript was not changed.

Direct solar radiation is replaced by downward solar radiation.

Table 3: Caption could be made a bit more informative. [...,...] indicates the range of a parameter that is calculated within the model. Presumably single values are where a fixed value is specified? Explain why several values depart from the control run values listed in section 2.2.

The caption was modified as follows:

Range of variation of model surface parameters for XP0. The values in square brackets indicate the values taken by a parameter when it is calculated by the model while the single values are fixed during the simulation.

Section 2.2 was rewritten and it is now explained that parameters are from the default model's configuration (cf answer to comment number 3)

Line 286: Change "Fig. 4 shows the evolution of the broadband albedos that vary over time. Indeed, the four models presented consider the variation of the albedo…" to "Fig. 4 shows the modeled broadband albedos in the four models that model the albedo…"

 done

Line 321: Change "radiation" to "energy".

done

Lines 389-390: "…that varies from simple to double" Please clarify

The text is changed into: that varies from 0.4 K (CHTESSEL) to 1 K (JULES).

Lines 390-392: "It should be noted that the single layer models (D95, CHTESSEL, EBA) have sometimes better results than the more sophisticated models which have to represent more physical processes, such as the evolution of albedo with time, the increase of snow density by compaction, among others." Maybe you could make this clearer on figure 8 by using a different shape of symbol for the multi-layer models?

Figure 8 was modified to distinguish between single-layer models and multilayer ones. Caption has been updated accordingly.

Line 442: "…gradient Richardson number…". Model parametrisations are usually based on a bulk Richardson number, calculated from the temperature difference between the lowest model level and the surface and the wind speed at the lowest model level. Include an equation that defines how the Richardson number is calculated.

The following text was added:

The bulk Richardson number is expressed as:

$$R_i = \frac{g}{\langle T \rangle} \frac{\Delta\theta \; \Delta z}{(\Delta U)^2} \qquad\qquad (7)$$

Where g is the acceleration of gravity, $\langle T \rangle$ the average virtual temperature, $\Delta\theta$ and $\Delta U$ the gradients of virtual potential temperature and wind speed of the considered layer of thickness $\Delta z$. The very low humidity of the air allows to assimilate the average virtual temperature and the virtual potential temperature to the average temperature and the average potential temperature.

---

## Author Comment (AC2)

**Author's response to RC2 comments**

**RC2 (Richard L.H. Essery)**

This is an interesting and generally well written paper, although it is unclear in parts. My comments are minor but numerous. The description of the simulation protocol needs to be improved; after reading the paper three times, I think there are three experiments – initial simulations, XP0 and XP1, also called "_new" – but I am rarely confident of which is being discussed at any time.

Thank you for the remark. The section 2.2 has been rewritten to make it clearer and better explain the protocol that has been used. Here is the new content of section 2.2:

The models were run offline, i.e. guided by the atmospheric forcing measured at Dome C, for a total simulation time of 15 days. Snow temperature was initialized from in situ measurements. First, each group had to provide the results obtained with the default settings of the surface parameters of their model. The models were run offline, i.e. guided by the atmospheric forcing measured at Dome C, for a total simulation time of 15 days. This set of simulations is called XP0, it includes one simulation for each of the 10 models and the name of an experiment is made up of the name of the model suffixed by "_xp0". From the simulations in the XP0 set, each participant could propose additional simulations. Only CHTESSEL and CLM4 performed calibration simulations, aiming at minimizing the root mean square error on the surface temperature. CLM4 calibrated the surface albedo and CHTESSEL calibrated the snowpack thickness and from this calibration then calibrated the dynamic and thermal roughness lengths. The names of the corresponding experiments are CLM4_cal1, CHTESSEL_cal1 and CHTESSEL_cal2 respectively. In addition, a rerun was proposed in order to better represent the diurnal cycle and to reduce the dispersion of the surface temperature results, but also to see if this dispersion was reduced or not in the coupled single-column simulations (this last point is not addressed in this study).

Snow albedo depends on zenith angle, but also on grain size and cloud cover. At Dome C, because the sky is generally clear, the effect of solar zenith angle is prominent compared to the typical diurnal cycle. Warren (1982) showed that the albedo of snow was maximum when the sun was low while its effect was less when the sun was at the zenith, it then enabled the surface to warm, or at least cool less by radiative effect. Most models do not consider the variation of albedo with the zenith angle, a fixed average value is proposed in the experimental protocol, corresponding to the average value of the ratio between the incident and reflected radiation measured at Dome C, over the period considered. Concerning the thermal emission of snow, the value of 0.98 is within the range of values commonly used for this type of medium. For the dynamic and thermal roughness lengths, values of 1 mm and 0.1 mm were chosen respectively. The dynamic roughness length is close to that established by Vignon et al. (2017a) who studied the effect of sastrugi on flow and momentum fluxes and proposed using a thermal roughness length that is one order of magnitude smaller than the dynamic roughness length. This ratio of 10 is classically used in many models calculating fluxes at the surface-atmosphere interface. Snow density at the surface can range from 20 kg m$^{-3}$ for fresh snow to 500 kg m$^{-3}$ for old, wet snow. Measurements at Dome C during the summer of 2014-2015 (Fréville, 2015) show that the snow density profile varies between 250 kg m$^{-3}$ and 310 kg m$^{-3}$ between the surface and 20 cm depth (Gallet et al., 2011).

Therefore, all participants were asked to run a new simulation with an albedo of 0.81, an emissivity of 0.98 (which corresponds to the average emissivity of the hemisphere (Armstrong and Brun 2008)), dynamic and thermal roughness lengths of 0.001 m and 0.0001 m, respectively, and for single-layer schemes, to impose a snow density of 300 kg m$^{-3}$, as well as the snow thermal coefficient $c_s$=3. 166 ×10$^{-5}$ (K m$^2$ J$^{-1}$). This coefficient is directly involved in the temperature evolution equation along the vertical:

$$C_{snow} \times \frac{\partial T(z,t)}{\partial t} = \frac{\partial}{\partial z}\left(\lambda(z)\frac{\partial T(z,t)}{\partial z}\right) \qquad (1)$$

$$c_s = (h_{snow} \times C_{snow})^{-1} \qquad (2)$$

Where $\lambda(z)$ is the heat conductivity of snow, $h_{snow}$ the snow depth (m) and $C_{snow}$ the volumetric heat capacity of the snow (J K$^{-1}$ m$^{-3}$).
Moreover,

$$C_{snow} = c_i \times \frac{\rho_{snow}}{\rho_i} \qquad (3)$$

where $c_i$ and $\rho_i$ are the heat capacity and density of the ice respectively. Combining equations (2) and (3) gives finally equation (4):

$$c_s = \rho_i \times (h_{snow} \times \rho_{snow} \times c_i)^{-1} \qquad (4)$$

Taking a thickness of $h_{snow} = 5$ cm, densities of snow and ice of 300 kg m$^{-3}$ and 900 kg m$^{-3}$ respectively, and the heat capacity of ice $c_i = 1.895 \times 10^6$ J K$^{-1}$ m$^{-3}$ we obtain according to equation (4) the value of $c_s$. This rerun is named XP1 and the name of the simulations that refer to it consists of the name of the model suffixed by "_xp1".
 Not all models were able to perform this new experiment, either due to lack of time or because the results came from an operational model that did not allow for adjustment of certain parameters or variables in the schemes. Although not all of them participated, it is interesting to study the impact of the changes induced by the XP1 configuration on the simulations of the XP0 ensemble, considering, when they exist, the calibrations. We therefore calculated the daytime and nighttime biases, as well as the difference in RMSD between XP1 and XP0 (or the simulation calibrated from a simulation of the XP0 ensemble), and evaluated the impact on the model error.

The introduction includes an extensive review of previous snow model intercomparisons, but these are all for seasonal snow with a focus on snowmelt and runoff, and are not very relevant for this paper. If going into this level of detail, however, uncertainty in model outputs due to uncertainty in meteorological driving data should also be mentioned

https://hess.copernicus.org/articles/19/3153/2015/hess-19-3153-2015.html

The text was modified as follows:

Previous snow model intercomparison have focused on seasonal snow with an emphasis on snowmelt and runoff, but here we are dealing with a different climate where snowfall is low in annual accumulation, and the snowpack is dry, subject to strong wind transport. However, a common feature with other intercomparison concerns the uncertainty in model outputs due to the uncertainty in the baseline meteorological data (Raleigh et al., 2015).

15

The abstract should acknowledge that the intercomparison is for a single site on the Antarctic Plateau

Done:

The results of offline numerical simulations, carried out during 15 days in 2009, on a single site on the Antarctic plateau, show that the simplest models are able to reproduce the surface temperature as well as the most complex models provided that their surface parameters are well chosen.

22

Can it be said that the surface temperature errors are consistent with the magnitude of sensible heat fluxes being too great both day and night?

A sentence was added in the abstract:

The surface temperature errors are consistent with too large a magnitude of sensible heat fluxes during the day and night.

27

This is a standard way to start a snow modelling paper, but snow is not a water resource at Dome C (and it isn't a key element of the landscape – it is the landscape!). The snow cover does not vary considerably in time and space.

I wanted to keep the beginning of the introduction as general as possible, so I kept it as is. However, I have added some elements to highlight the specificities of the Dome C site.

In Antarctica, and more particularly in the interior of the continent, as at Dome Charlie (Dome C hereafter), conditions are very different, since snow is the landscape, and it shows relatively little spatial and temporal variation. In addition, the conditions are very dry and cold, which prevents the snow from melting and precipitation is rare. At Dome C, the small amount…

69

"snow patterns" suggest spatial distribution of snow cover, which is not considered in Schlosser et al. (2000).

Snow patterns was changed into snow model complexity

88

There are two sites in Etchevers et al. (2004).

a snow-covered low vegetation site was changed into two mountainous alpine sites

133

"that that water may freeze"

done (one "that" was removed)

150

The information in Table 1 is already provided in the text and the author list; it could be deleted.

done

154

All of the information in this sentence is repeated with more detail in the following sentences.

Section 2.2 was rewritten.

173

Rather than a list in the text, layer depths might be better presented in a table, which could then include the initial temperature and density profiles. How was initial temperature prescribed for single-layer models?

This information has been added to the end of the new section 2.2 and a table was added:

For the multilayer models, the snow density and temperature profiles were initialized from observations. Note that the single-layer models use a fixed density close to 300 kg m$^{-3}$ which corresponds to a depth of about 10 cm in the initial profile, and their initial temperature was also provided from in situ measurements. Table 1 describes the vertical discretization and gives the initial temperature and snow density profiles. The LMDZ model is a special case. Indeed, it is a ground thermal model with the thermal inertia of snow that is used and not really a snow scheme, which is why there is no snow density as such.

174

The sensitivity tests mentioned here are XP0 and XP1, not additional test deemed relevant by the participants?

No only 2 models performed calibration tests to minimize the surface temperature RMSD. It is explained in section 2.2.

179

No need for equations 1 and 2 to be bracketed with {

done

196

This is the only mention of calibration. It is important to know if some of the models have been calibrated for these simulations, and how.

It is now explained in section 2.2

205

The K&Z CM22 does not just measure visible radiation (and, in the Figure 1 caption, not just direct solar radiation).

The table caption has been changed to explain that the focus is on the forcing data and indicates which sensor was used to measure it (this does not mean that the sensor does not measure something else):

Presentation of the forcing data (name, unit and position on the measuring mast) and instruments used for the measurements.

Direct solar was changed into downward solar in Fig. 1

Table 2

Wind direction is not used. A Vaisala HMP155 measures relative humidity, which requires temperature for conversion to specific humidity. Genthon et al. (2017) consider humicap measurements to be biased low for the conditions of Dome C.

https://acp.copernicus.org/articles/17/691/2017/acp-17-691-2017.pdf

Wind direction was removed from table 1. Reference to HMP155 for air temperature was removed.

245

Could say when the temperature probes were installed.

It is now indicated in the text (26 November 2006)

Figure 2

Why are temperatures shown as not filled at four depths?

It was an error and the figure has been updated

Table 3

The caption should explain the use of square brackets.

The caption was modified into:

Table 2: Range of variation of model surface parameters for XP0. The values in square brackets indicate the values taken by a parameter when it is calculated by the model while the single values are fixed during the simulation.

316

I assume this is not a linear profile between the surface and 10 m depth.

Right, see answer to item 173 where a vertical description is given.

Figure 5

Adding air temperature to this figure would be an interesting comparison.

Done and the caption was updated.

341

The PDFs in Figure 6 have too many extrema to be cubic functions. Were they, in fact, fitted with cubic splines?

Functions was replaced by splines

389

What does "a root mean square error that varies from simple to double" mean?

The text was modified into:

a root mean squared error that varies from 0.4 K (CHTESSEL) to 1 K (JULES)

391

The more sophisticated model do not have to represent the evolution of albedo in XP0, as I understand it.

No, they do have to model it, in fact the albedo parameterization is not disactivated. The protocol is better explained in section 2.2.

400

Wind speed is always greater than or equal to zero, so taking its modulus does not add anything.

"Modulus of the" was removed from this sentence

401

The proportionality constant is not simply the surface exchange coefficient; air density, and heat capacity for sensible heat flux, are also required.

The sentence was replaced by:

For sensible heat flux, the proportionality coefficient is the turbulent surface exchange coefficient multiplied by the air density and the heat capacity.

417

Assumed overestimation of sensible heat fluxes in stable conditions is a longstanding feature of models, although it can prevent larger biases in surface temperature.

https://journals.ametsoc.org/view/journals/clim/10/6/1520-0442_1997_010_1273_votseb_2.0.co_2.xml

This comment and the reference were added in the text:

The assumed overestimation of sensible heat fluxes under stable conditions is a long-standing feature of the models, although it may prevent larger biases in the surface temperature (King and Connolley, 1997).

430

Equation 5 is how Qh is measured and equation 6 is how it is modelled.

Measured and modelled by were added in the text

442

This is the bulk Richardson number, not the gradient Richardson number. I would have guessed that GDPS and CLM4 are singled out because they characterize stability by the Obukhov length, but that is the case for JULES also.

Gradient Richardson number was changed into bulk Ri. Yes, that's right GDPS4 and CLM4 are singled because they characterize stability in a different manner than the other models using LMO. But that is not the case for JULES which is also using the Louis model based on Ri.

464

The vertical temperature profile is well initialized by construction.

Right, I wanted to mention that the initialization was identical and correctly set up.

The sentence "It can be seen that the vertical temperature profile is well initialized for the four models." was changed into:

It can be seen that the initialization of the vertical temperature profile is identical and correctly configured for all four models.